# Recent African strains of Zika virus display higher transmissibility and fetal pathogenicity than Asian strains

Fabien Aubry[1,16], Sofie Jacobs[2,16], Maïlis Darmuzey[3,16], Sebastian Lequime[4,5], Leen Delang[2], Albin Fontaine[6,7,8], Natapong Jupatanakul[1,9], Elliott F. Miot[1], Stéphanie Dabo[1], Caroline Manet[10], Xavier Montagutelli[10], Artem Baidaliuk[1,11], Fabiana Gámbaro[11], Etienne Simon-Lorière[11], Maxime Gilsoul[3], Claudia M. Romero-Vivas[12], Van-Mai Cao-Lormeau[13], Richard G. Jarman[14], Cheikh T. Diagne[15], Oumar Faye[15], Ousmane Faye[15], Amadou A. Sall[15], Johan Neyts[2], Laurent Nguyen[3], Suzanne J. F. Kaptein[2✉] & Louis Lambrechts[1✉]

The global emergence of Zika virus (ZIKV) revealed the unprecedented ability for a mosquito-borne virus to cause congenital birth defects. A puzzling aspect of ZIKV emergence is that all human outbreaks and birth defects to date have been exclusively associated with the Asian ZIKV lineage, despite a growing body of laboratory evidence pointing towards higher transmissibility and pathogenicity of the African ZIKV lineage. Whether this apparent paradox reflects the use of relatively old African ZIKV strains in most laboratory studies is unclear. Here, we experimentally compare seven low-passage ZIKV strains representing the recently circulating viral genetic diversity. We find that recent African ZIKV strains display higher transmissibility in mosquitoes and higher lethality in both adult and fetal mice than their Asian counterparts. We emphasize the high epidemic potential of African ZIKV strains and suggest that they could more easily go unnoticed by public health surveillance systems than Asian strains due to their propensity to cause fetal loss rather than birth defects.

[1] Insect-Virus Interactions Unit, Institut Pasteur, UMR2000, CNRS, Paris, France. [2] KU Leuven Department of Microbiology, Immunology and Transplantation, Rega Institute for Medical Research, Laboratory of Virology and Chemotherapy, Leuven, Belgium. [3] GIGA-Stem Cells/GIGA-Neurosciences, Interdisciplinary Cluster for Applied Genoproteomics (GIGA-R), C.H.U. Sart Tilman, University of Liège, Liège, Belgium. [4] KU Leuven Department of Microbiology, Immunology and Transplantation, Rega Institute, Laboratory of Clinical and Epidemiological Virology, Leuven, Belgium. [5] Cluster of Microbial Ecology, Groningen Institute for Evolutionary Life Sciences, University of Groningen, Groningen, The Netherlands. [6] Unité Parasitologie et Entomologie, Département Microbiologie et Maladies Infectieuses, Institut de Recherche Biomédicale des Armées (IRBA), Marseille, France. [7] IRD, SSA, AP-HM, UMR Vecteurs—Infections Tropicales et Méditerranéennes (VITROME), Aix Marseille University, Marseille, France. [8] IHU Méditerranée Infection, Marseille, France. [9] National Center for Genetic Engineering and Biotechnology (BIOTEC), Pathum Thani, Thailand. [10] Mouse Genetics Laboratory, Institut Pasteur, Paris, France. [11] Evolutionary Genomics of RNA Viruses Group, Institut Pasteur, Paris, France. [12] Laboratorio de Enfermedades Tropicales, Departamento de Medicina, Fundación Universidad del Norte, Barranquilla, Colombia. [13] Institut Louis Malardé, Papeete, Tahiti, French Polynesia. [14] Viral Diseases Branch, Walter Reed Army Institute of Research, Silver Spring, MD, USA. [15] Arbovirus and Viral Hemorrhagic Fevers Unit, Institut Pasteur Dakar, Dakar, Senegal. [16] These authors contributed equally: Fabien Aubry, Sofie Jacobs, Maïlis Darmuzey. ✉email: suzanne.kaptein@kuleuven.be; louis.lambrechts@pasteur.fr

Zika virus (ZIKV) is a flavivirus mainly transmitted among humans through the bite of infected *Aedes aegypti* mosquitoes[1,2]. After its first isolation from a sentinel monkey in Uganda in 1947, ZIKV was shown to circulate in enzootic sylvatic cycles in Africa and continental Asia, but human infections were only sporadically reported for half a century[3–5]. The first documented human epidemic of ZIKV occurred in 2007 on the Pacific island of Yap, Micronesia[6]. Subsequent larger ZIKV outbreaks were recorded in French Polynesia and other South Pacific islands in 2013–2014 (refs. [7,8]). In May 2015, ZIKV was detected for the first time in Brazil from where it rapidly spread across the Americas and the Caribbean, causing an epidemic of unprecedented magnitude involving hundreds of thousands of human cases[9]. Whereas human ZIKV infections are usually asymptomatic or result in a self-limiting mild illness, ZIKV was associated for the first time with severe neurological complications such as Guillain-Barré syndrome (GBS) in adults, and congenital Zika syndrome (CZS), a spectrum of fetal abnormalities and developmental disorders including microcephaly, when mothers were infected during early pregnancy[10,11]. Within less than a decade, ZIKV went from a poorly known virus causing sporadic human infections in Africa and Asia to a nearly pandemic neurotropic virus with active circulation detected in more than 87 countries and territories[9]. Phylogenetic analyses of ZIKV genetic diversity identified two major ZIKV lineages referred to as the African lineage and the Asian lineage, respectively[12]. Strikingly, all ZIKV strains responsible for human outbreaks to date belong to the Asian lineage[9,13].

The explosiveness and magnitude of worldwide ZIKV emergence increased awareness and surveillance in regions with seemingly favorable conditions, such as Asia or Africa. Retrospective analyses of samples and surveillance programs in several Asian countries revealed that ZIKV had circulated at low but sustained levels for decades[14,15]. Improved case recognition shed light on small outbreaks in Singapore[16], Vietnam[17] and India[18] and led to the first reports of birth defects caused by indigenous ZIKV strains in South East Asia[19–22]. In Africa, where both ZIKV and *Ae. aegypti* mosquitoes are present, only one human outbreak was reported in the archipelago of Cape Verde between 2015 and 2017 (ref. [23]). Autochthonous ZIKV transmission was also detected in Angola during the same time period with four confirmed acute Zika cases and several suspected cases of microcephaly. Phylogenetic analyses revealed that the ZIKV strains detected in Cape Verde and Angola belonged to the Asian lineage and were probably independently imported from Brazil[24,25]. So far, the African ZIKV lineage has never been detected outside the African continent and never been associated with epidemic transmission, birth defects or neurological disorders[9,13].

Surprisingly, a growing body of experimental evidence, both in vitro and in vivo, points towards a higher transmissibility and pathogenicity of the African ZIKV strains compared to their Asian counterparts[13]. African ZIKV strains typically cause more productive and more lethal infections than Asian strains in cell culture[26–32], they are more transmissible by mosquitoes[33–37] and they are associated with more severe pathology in adult mice and mouse embryos[32,38–46]. A few studies, however, reported evidence supporting the opposite conclusion in nonhuman primates[47–49], various cell types[50,51] and mosquitoes[45]. This discrepancy may reflect the lack of standard panels of ZIKV strains and/or the scarcity of recent African ZIKV strains available from public biobanks and laboratory collections[52]. Indeed, most of the available African ZIKV strains were isolated several decades ago and often underwent numerous passages in cell culture and/or suckling mouse brains[53], questioning their biological relevance for comparative studies and experimental assessments of their epidemic potential.

To more rigorously assess the relative epidemic potential of the Asian and African ZIKV lineages, we compared their transmissibility by mosquitoes and pathogenicity in immunocompromised mice using a panel of seven recent, low-passage ZIKV strains representing the current viral genetic diversity. Using the newly generated empirical data and a previously described stochastic agent-based model[54], we performed outbreak simulations in silico to quantify the epidemiological dynamics of each ZIKV strain. Finally, we used a mouse model of ZIKV-induced microcephaly to evaluate the ability of the ZIKV strains to disrupt embryonic development in utero.

## Results

To perform a comprehensive phenotypic characterization in both mosquito and mouse models, we assembled a set of seven recently isolated ZIKV strains based on their broad phylogenetic coverage, worldwide geographical distribution, and minimal passage history. Our ZIKV panel included two recent strains from the African lineage (Senegal_2011; Senegal_2015), three non-epidemic strains from the Asian lineage (Philippines_2012, Cambodia_2010, Thailand_2014), and two epidemic strains from the Asian lineage (F_Polynesia_2013, Puerto_Rico_2015) (Table S1; Fig. 1).

**African ZIKV strains are more transmissible by mosquitoes than Asian strains.** To evaluate variation in transmissibility by wild-type *Ae. aegypti* between the ZIKV strains, we examined the mosquito infection rate (proportion of blood-fed mosquitoes with body infection, determined by RT-PCR) and transmission efficiency [proportion of blood-fed mosquitoes with infectious saliva, determined by focus-forming assay (FFA)] following exposure to an artificial infectious blood meal. We monitored infection rate and transmission efficiency from day 7 to day 17 post oral exposure because transmission rarely occurs and infection rates can be underestimated at earlier time points[55]. In a first experiment, mosquitoes were orally exposed to a relatively high dose of the five Asian ZIKV strains and of the Senegal_2015 strain. Experimental variation in infectious dose was minimal between the ZIKV strains, ranging from 5.6 to 5.8 $\log_{10}$ focus-forming units (FFU) per ml of artificial blood meal. Based on a typical blood meal size of 2.5 µl[56], this corresponded to an ingested dose ranging from 995 to 1577 FFU per *Ae. aegypti* female. The mosquito infection rates were consistently high, ranging from 82 to 100% across ZIKV strains and time points (Fig. 2a). They did not differ statistically at any of the time points with the exception of day 7 (logistic regression: $p = 0.0155$). In contrast, the transmission efficiency of the Senegal_2015 strain was significantly higher at all time points (logistic regression: $p = 0.0224$ on day 7 and $p < 0.0001$ at later time points), reaching 83% of infectious mosquitoes at the end of the time course (Fig. 2b). Infectious viral particles were detected in mosquito saliva as early as 7 days post blood meal for the Senegal_2015 strain and only after 10 days for the Cambodia_2010 strain, 14 days for the Philippines_2012, F_Polynesia_2013 and Puerto_Rico_2015 strains, and 17 days for the Thailand_2014 strain. Final transmission efficiency also differed between Asian strains, ranging from 10% for the Thailand_2014 strain to 50% for the F_Polynesia_2013 strain on day 17 post blood meal.

Next, we tested whether the superior transmissibility of the Senegal_2015 strain was representative of the African ZIKV lineage or specific to this strain. We conducted a second experiment that included the two African ZIKV strains of the panel and the F_Polynesia_2013 strain, which was the best-transmitted Asian strain in the first experiment. To avoid saturation and increase our ability to detect differences in

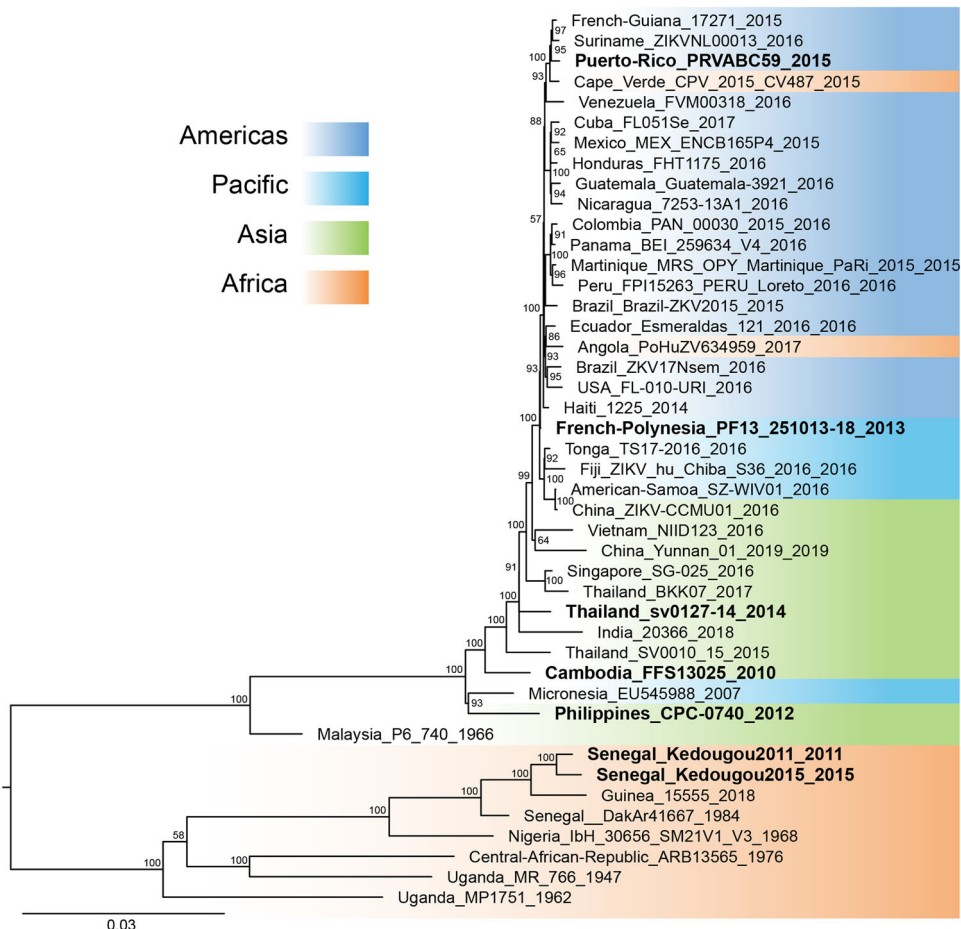

**Fig. 1 Phylogenetic position of ZIKV strains used in this study.** The phylogenetic tree shows the seven ZIKV strains of the panel (in bold) among a backdrop of ZIKV strains spanning the current viral genetic diversity. The colored background represents the geographic origin of ZIKV strains. The consensus tree was generated from 1000 ultrafast bootstrap replicate maximum-likelihood trees, using a GTR + F + G4 nucleotide substitution model of the full ZIKV open reading frame. The tree is midpoint rooted and the root position is verified by the Spondweni virus outgroup on amino-acid and codon-based trees. Support values next to the nodes indicate ultrafast bootstrap proportions (%) and the scale bar represents the number of nucleotide substitutions/site.

infection rate, we used a lower infectious dose (4.7–4.8 $\log_{10}$ FFU/ml) than in the first experiment. Based on a typical blood meal size of 2.5 µl[56], this corresponded to an ingested dose ranging from 125 to 158 FFU per mosquito. The infection rates remained relatively high (68–93%) for the two African strains, whereas it was significantly lower at all time points for the F_Polynesia_2013 strain (logistic regression: $p = 0.0384$ on day 17 and $p < 0.0001$ at earlier time points), increasing from 24% on day 7 to 59% on day 17 (Fig. 2c). The difference was more striking (logistic regression: $p = 0.1086$ on day 7 and $p < 0.0003$ at later time points) for the transmission efficiency (Fig. 2d). Between day 7 and day 17, transmission efficiency of the Senegal_2015 and Senegal_2011 strains increased from 0 to 52% and from 7 to 70%, respectively, whereas no infectious particles were detected in any of the saliva samples collected from mosquitoes infected with the F_Polynesia_2013 strain throughout the time course. These results indicate that ZIKV strains of the African lineage, in general, display a significantly higher transmission potential than ZIKV strains of the Asian lineage.

To translate the observed variation in transmissibility between ZIKV strains into differences in epidemic risk, we incorporated our empirical data into a stochastic agent-based model to perform outbreak simulations in silico. Mosquito-to-human ZIKV transmission events in the simulations were governed by log-logistic

regression parameters (Table S2) estimated from the ZIKV strain-specific data obtained in the high-dose experiment described above (Fig. 2a, b). Human-to-mosquito transmission events depended on shared parameters among the ZIKV strains, which were derived from an independent experiment in which batches of naïve mosquitoes from Guadeloupe were allowed to feed daily on ZIKV-infected mice (Cambodia_2010 strain) during the course of their viremic period (Fig. S1). The magnitude of the outbreak was defined according to the number of secondary infections in a simulated population of 100,000 humans, ranging from a lack of outbreak (no secondary infection), to small-scale outbreaks (<100 secondary infections) and to large-scale outbreaks (≥100 secondary infections). All ZIKV strains were able to cause secondary human infections, however the risk and magnitude of the outbreak greatly varied among the strains (Fig. 3). The proportion of simulations resulting in at least one secondary human infection ranged from 21% (Puerto_Rico_2015) to 63% (Senegal _2015). The proportion of small-scale outbreaks ranged from 2% (Senegal_2015) to 23% (Thailand_2014) and the proportion of large-scale outbreak ranged from 9% (Thailand_2014) to 61% (Senegal_2015). We did not observe a clear association between the epidemic or non-epidemic status of the ZIKV strains and the probability and magnitude of outbreaks. Using a less susceptible mosquito population from Gabon to model human-to-mosquito transmission events

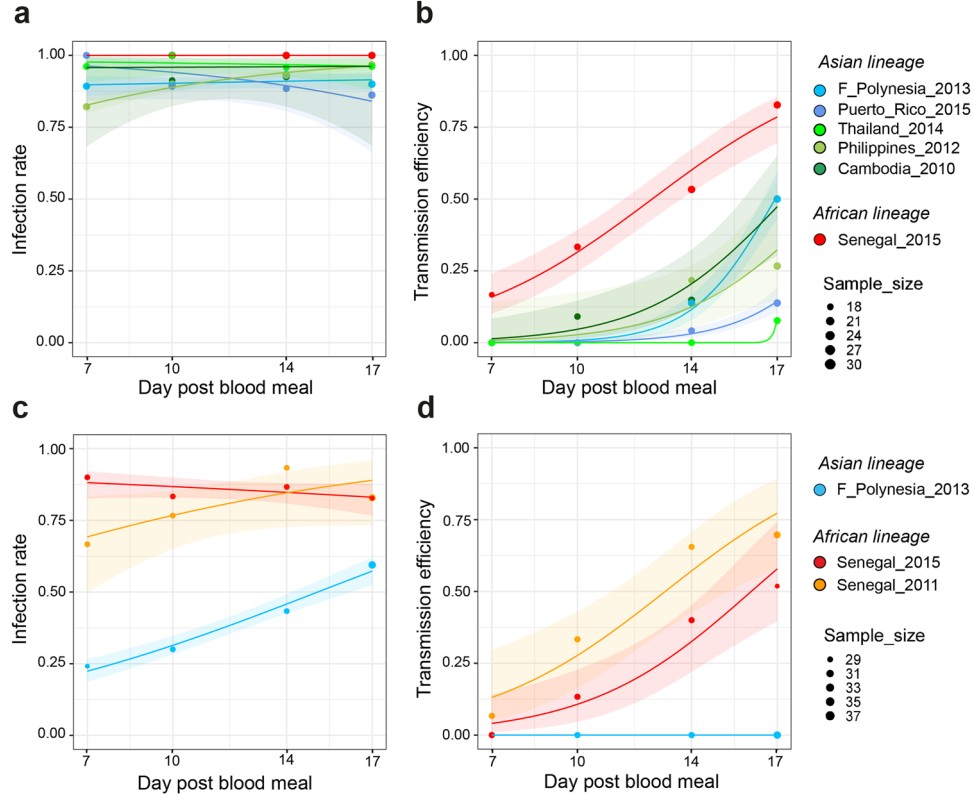

**Fig. 2 Mosquito infection rate and transmission efficiency of African and Asian ZIKV strains.** Wild-type *Ae. aegypti* mosquitoes from Colombia were orally exposed to ZIKV and collected on day 7, 10, 14, and 17 post infectious blood meal to analyze their carcasses and saliva samples collected in vitro. Infection rates and transmission efficiencies over time are shown for each ZIKV strain tested after oral exposure to a high dose (5.6–5.8 log$_{10}$ FFU/ml) (**a**, **b**) or a low dose (4.7–4.8 log$_{10}$ FFU/ml) (**c**, **d**) of virus. Infection rate is the proportion of ZIKV-positive carcasses among all blood-fed mosquitoes (determined by RT-PCR). Transmission efficiency is the proportion of blood-fed mosquitoes with infectious saliva (determined by FFA). The data points represent the empirically measured proportions, and their size is proportional to the sample size (high dose: n = 18–30 mosquitoes per group; low dose: n = 29–37 mosquitoes per group). The lines represent the logistic regression results and the shaded areas are the 95% confidence intervals of the logistic fits. Source data are provided as a Source Data file.

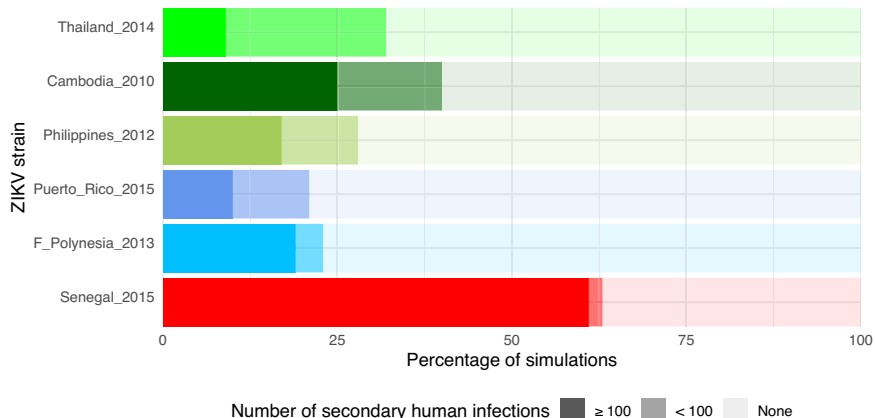

**Fig. 3 Simulated effect of empirical variation in ZIKV transmissibility on the risk and magnitude of human outbreaks.** A stochastic agent-based model was run 100 times based on the experimentally determined kinetics of mosquito transmission of six ZIKV strains. Other parameters of the model, such as the mosquito biting rate and infection dynamics within the human host, were shared between viruses. The figure shows the proportion of simulated outbreaks that resulted in ≥100, <100, and no secondary human infections.

(Fig. S1) resulted in reduced epidemic potential for all ZIKV strains (Fig. S2). Together, the simulation results indicate that the higher transmissibility of the African ZIKV strains in our laboratory experiments translates into a markedly higher probability and size of human outbreaks predicted by our epidemiological model.

**African ZIKV strains are more lethal than Asian strains in immunocompromised adult mice.** To assess the ability of ZIKV strains to cause more or less severe disease in a mammalian model, we inoculated AG129 mice with 10$^3$ plaque-forming units (PFU) of ZIKV and monitored their viremia, body weight, and clinical signs of disease. Viremia levels ranged from 4.71 to 9.15

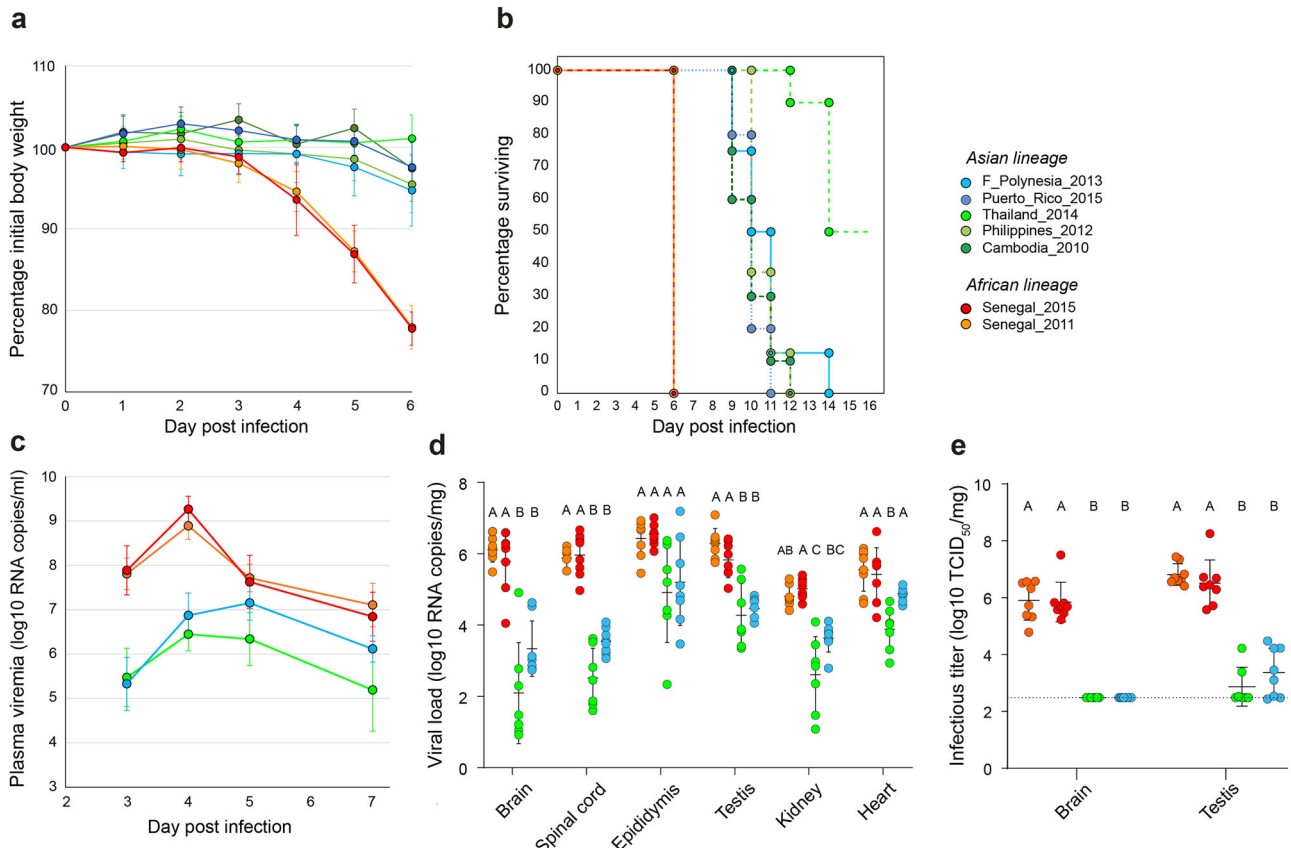

**Fig. 4 Pathogenicity of African and Asian ZIKV strains in immunocompromised adult mice.** In a first experiment (**a**, **b**), male AG129 mice were inoculated with $10^3$ PFU of ZIKV. Each virus strain was represented by $n = 10$ mice, with the exception of the F_Polynesia_2013 and Philippines_2012 strains that were represented by $n = 8$ mice. **a** Mouse weight over time is shown as the percentage of body weight prior to infection (mean ± standard error). **b** Mouse survival over time is shown as the percentage of mice alive. Mice were euthanized when reaching humane endpoints (weight loss >20% or/and severe symptom onset). In a second experiment (**c**–**e**), male AG129 mice were inoculated with 1 PFU of ZIKV ($n = 8$ mice per strain). **c** Time course of mouse viremia is shown in $\log_{10}$-transformed viral genome copies per ml of plasma (mean ± standard error). Three extreme outliers were excluded for the Senegal_2015 strain. **d** Viral loads in organs collected on day 7 post infection are shown in $\log_{10}$-transformed viral genome copies per mg of tissue. Statistical significance of differences was determined by one-way ANOVA followed by Tukey's post hoc test for brain, heart and testis, by Brown–Forsythe and Welch ANOVA followed by Games-Howell's post hoc test (two sided) for epididymis and spinal cord, and by Kruskal–Wallis rank-sum test followed by Dunn's post hoc test (two sided) for kidney. Viral loads were significantly higher for African than for Asian ZIKV strains in the brain ($p < 0.0001$), spinal cord ($p < 0.0001$), testis ($p < 0.0001$), kidney ($p < 0.0001$), and heart ($p < 0.0001$). **e** Infectious virus in brain and testis collected on day 7 post infection are shown in $\log_{10}$-transformed 50% tissue-culture infectious dose ($TCID_{50}$) per mg of tissue. The horizontal dotted line indicates the lower limit of detection of the assay (310 $TCID_{50}$ units per mg of tissue). Statistical significance of differences was determined by Kruskal–Wallis rank-sum test followed by Steel-Dwass's post hoc test for brain and by one-way ANOVA followed by Tukey's post hoc test for testis. Infectious titers were significantly higher for African than for Asian ZIKV strains in the brain ($p < 0.0001$) and testis ($p < 0.0001$). In (**d**, **e**), data are presented as mean ± standard deviation and ZIKV strains not sharing a letter above the graph are statistically significantly different ($p < 0.05$). Source data are provided as a Source Data file.

$\log_{10}$ viral RNA copies/ml of plasma and peaked on day 3 post infection except for the Philippines_2012 strain, for which viremia peaked on day 2 (Fig. S3). Across the viremic period, the average viremia level was 7.04 $\log_{10}$ viral RNA copies/ml for the African ZIKV lineage and 6.60 $\log_{10}$ viral RNA copies/ml for the Asian ZIKV lineage. Accounting for differences between strains within each ZIKV lineage and the random effect of individual mice, we found that the kinetics of viremia differed significantly between the Asian and the African ZIKV lineages (repeated measures analysis: $p < 0.0001$). The viremia levels of African ZIKV strains increased with a lag but to higher levels during days 1–3 post infection, decreased faster during days 4–5 post infection, and increased again during days 5–6 post infection, relative to the Asian strains (Fig. S3). During the first 6 days of infection, the average body weight of mice infected with Asian ZIKV strains was 100.2% (range: 85.8–109.0%) of their initial

weight, however it was 92.8% (range: 74.4–104.7%) for mice infected with African ZIKV strains (Fig. 4a). Accounting for differences between strains within each ZIKV lineage and the random effect of individual mice, the kinetics of body weight differed significantly between the Asian and the African ZIKV lineages (repeated measures analysis: $p < 0.0001$). Mice infected with African ZIKV strains lost significantly more weight than mice infected with the Asian strains during days 3–6 post infection (Fig. 4a). Following infection, mouse survival differed significantly between ZIKV strains (log-rank test: $p < 0.0001$). All mice infected with the African ZIKV strains became morbid and reached humane endpoints on day 6 post infection (Fig. 4b). In contrast, mice infected with the Asian strains developed disease symptoms significantly later and only started to reach humane endpoints from day 9 post infection onwards. Their median time to death ranged from 10 to 10.5 days, with the exception of

the Thailand_2014 strain for which 50% of the mice were still alive at the end of the follow-up period (Fig. 4b). These results show that both of our African ZIKV strains are more pathogenic overall than their Asian counterparts in immunocompromised adult mice.

AG129 mice are highly permissive to ZIKV[57] and viremia levels may thus easily saturate when they are inoculated with a high dose of virus. To avoid saturation and enhance our ability to detect differences in viral RNA levels in plasma and tissues, and to delay the onset of disease in mice infected with the African ZIKV strains, we performed another experiment with a 1000-fold lower inoculum (1 PFU). This experiment included both African ZIKV strains and two Asian ZIKV strains recapitulating the variation seen in the previous experiment. The Thailand_2014 strain was chosen as a pre-epidemic strain displaying the most attenuated phenotype and the epidemic F_Polynesia_2013 strain represented the other Asian ZIKV strains. Surprisingly, using a 1000-fold lower inoculum delayed the onset of disease by only 1 day for the African ZIKV strains, for which all mice had to be euthanized on day 7 post infection. This result clearly highlighted the higher pathogenicity of the African ZIKV lineage. To enable a proper comparison of viral RNA levels in mouse tissues between ZIKV strains, all the other mice were also euthanized on day 7 post infection to collect their organs. Lowering the inoculum delayed the peak of viremia in all ZIKV-infected mice, which occurred on day 5 post infection for the F_Polynesia_2013 strain and on day 4 for the other ZIKV strains (Fig. 4c). The level of plasma viremia was overall significantly higher (repeated measures analysis: $p < 0.0001$) and decreased more sharply during days 4–5 post infection ($p < 0.0001$) for the African ZIKV strains than for the Asian ZIKV strains (Fig. 4c). Likewise, viral RNA levels measured in organs collected on day 7 post infection were consistently higher for the mice infected with the African ZIKV strains (Fig. 4d). African ZIKV strains resulted in significantly higher viral loads than Asian strains in the brain ($p < 0.0001$), spinal cord ($p < 0.0001$), testis ($p < 0.0001$), kidney ($p < 0.0001$), and heart ($p < 0.0001$). We confirmed that differences in viral RNA levels translated into differences in infectious titers in brain and testis samples, for which sufficient material was available to also perform endpoint titrations. Indeed, we found that African ZIKV strains were associated with significantly higher levels of infectious virus in the brain ($p < 0.0001$) and testis ($p < 0.0001$) of infected mice than Asian strains (Fig. 4e). These results indicate that whereas ZIKV strains from both lineages can cause systemic infections with similar organ tropism in this mouse model, the African strains are more pathogenic and result in significantly higher morbidity and mortality.

**African ZIKV strain causes fetal death in immunocompetent mice.** To investigate differences in vertical transmission phenotypes between ZIKV strains, we used a recent model of ZIKV-induced microcephaly by intraplacental injection in mouse embryos[58,59]. We first performed intraplacental injections of the Senegal_2015, Thailand_2014 and F_Polynesia_2013 ZIKV strains into the labyrinth of SWISS mouse embryos at E10.5 and compared the infection outcomes at E14.5. We observed subcutaneous edema in E14.5 embryos 4 days after intraplacental ZIKV injection for all strains (Fig. S4). Subcutaneous edema was significantly more frequent ($p < 0.0012$) in embryos infected with the Senegal_2015 strain (91%; $n = 11$) than in embryos infected with the Thailand_2014 strain (30%; $n = 16$), the F_Polynesia_2013 strain (6%; $n = 16$), or mock-injected embryos (0%; $n = 10$). We next compared the extent of ZIKV systemic infection in E14.5 embryos following intraplacental injection at

E10.5. ZIKV immunolabeling showed a comparable distribution of all ZIKV strains in brain, lung, heart, liver, intestine, eye, spinal cord and atriums of infected embryos (Figs. 5a, S5). The overall immunofluorescence staining was stronger for the Senegal_2015 and Thailand_2014 strains relative to the F_Polynesia_2013 strain. We confirmed these observations quantitatively by measuring viral RNA levels in the brain, lung, heart, liver and intestine. In all organs, viral loads differed significantly between ZIKV strains ($p < 0.0277$), with the F_Polynesia_2013 strain consistently resulting in significantly lower viral loads than the Senegal_2015 strain, as well as the Thailand_2014 strain in most organs (Fig. 5b).

To measure the impact of different ZIKV strains on embryonic brain development, we performed intraplacental injections at E10.5 and examined embryos at E18.5. Injection of the Senegal_2015 strain caused massive resorption resulting in the death of all infected embryos harvested at E18.5 (Fig. 6a). In contrast, infection with the Asian ZIKV strains (F_Polynesia_2013 and Thailand_2014) was not lethal to E18.5 embryos but resulted in a significant reduction ($p < 0.05$) in head weight (Fig. 6c), cortical thickness (Fig. 6b, e) and number of cortical cells (Fig. 6f) compared to the mock-injected embryos. In addition, we detected a significant reduction in brain weight (Fig. 6d) and ventriculomegaly (Fig. 6b, g) with the Thailand_2014 strain but not the F_Polynesia_2013 strain. Together, these results show that ZIKV strains with higher levels of infection at E14.5 are also associated with more severe phenotypes at E18.5. The Senegal_2015 strain caused embryonic death before E18.5 and the Thailand_2014 strain resulted in more pronounced microcephaly and ventriculomegaly than the F_Polynesia_2013 strain.

## Discussion

By comparing seven ZIKV strains representing the current breadth of viral genetic diversity worldwide, this study provides clear experimental evidence that recent African strains are more transmissible and potentially more pathogenic than Asian strains. In our experiments, ZIKV strains of the African lineage were more infectious to and were transmitted faster by wild-type *Ae. aegypti* mosquitoes from Colombia, translating into a higher epidemic potential in outbreak simulations. In addition, ZIKV strains of the African lineage were more pathogenic to immunocompromised adult mice and caused massive resorption and embryonic death in immunocompetent mouse embryos infected in utero by intraplacental injection.

Assessing ZIKV pathogenicity in the vertebrate host is complicated by the limited number of animal models that are available. Nonhuman primate infections closely emulate human infections but they raise ethical issues and are generally restricted to vaccine and drug development[60]. Several models of ZIKV pathogenesis in adult mice have been developed that recapitulate various features of human disease[38,61,62]. In general, wild-type mice can be infected with ZIKV but they do not develop overt clinical illness and little or no viremia[38]. In contrast, mice lacking the ability to produce or respond to type I interferon typically develop severe neurological disease associated with high viral loads in key organs and substantial lethality. We used immunocompromised AG129 mice (deficient in interferon α/β and γ receptors) as a convenient proxy to assess pathogenesis in our panel of ZIKV strains. These mice are very susceptible to ZIKV infection[57], making them highly suitable to monitor viral kinetics and disease manifestations. ZIKV strains of the African lineage caused significantly more morbidity and mortality than did their Asian counterparts, despite a similar tissue tropism. Of note, the level of pathogenicity observed in the immunocompromised adult

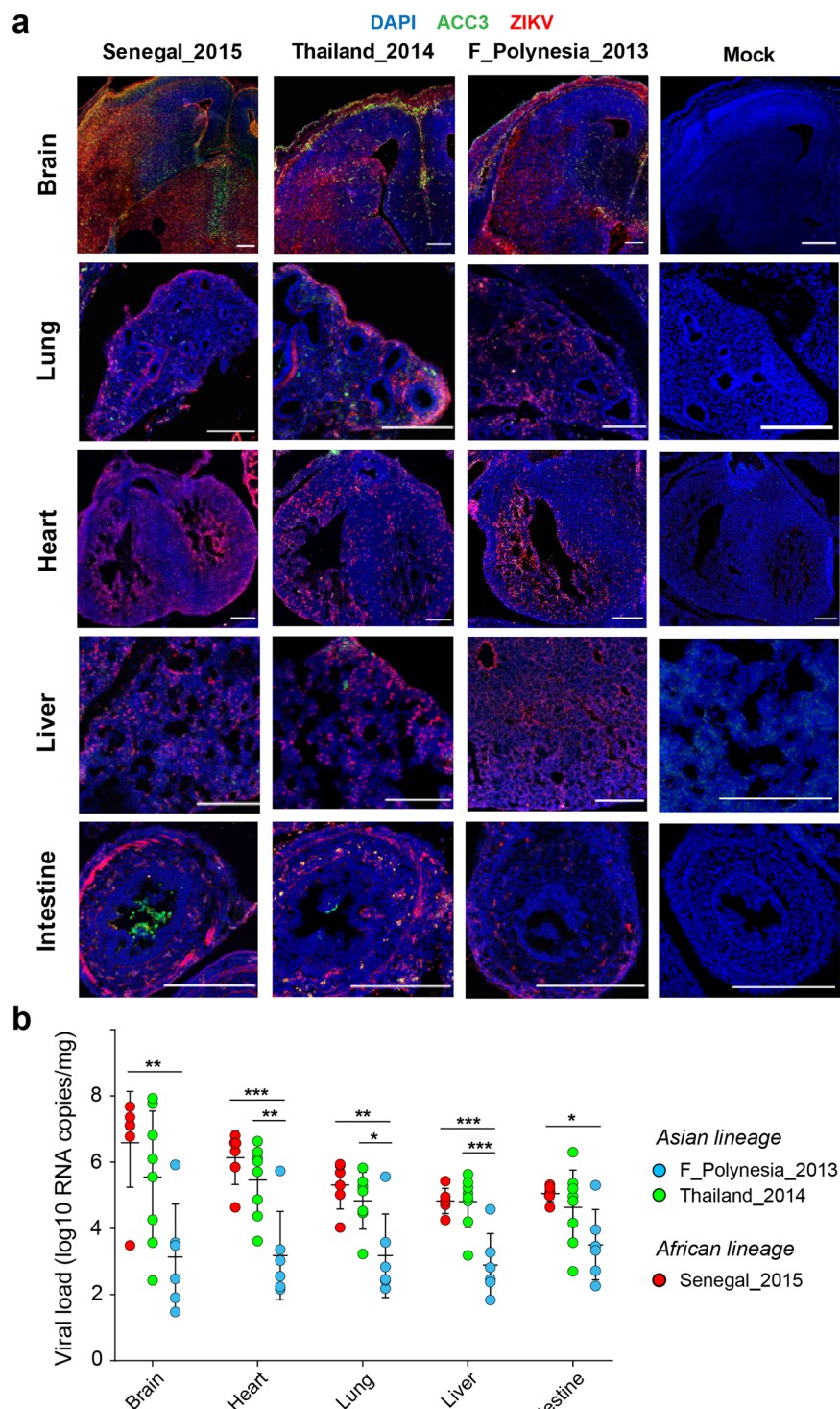

**Fig. 5 Organ tropism and viral load of African and Asian strains of ZIKV in vertically infected mouse embryos.** Immunocompetent mouse embryos were infected at E10.5 by intraplacental injection of 500–1000 PFU of ZIKV and analyzed at E14.5 by microdissection. **a** Immunolabeling of embryonic brain, lung, heart, liver and intestine sections representative of each ZIKV strain tested ($n = 3$ embryos per strain). Blue, green and red colors indicate DAPI, anti-cleaved caspase 3 (ACC3) and ZIKV stainings, respectively. The scale bars represent 200 μm. **b** Viral load of embryonic brain, lung, heart, liver and intestine are shown for each ZIKV strain in viral genome copies per organ. Data are presented as mean ± standard deviation and represent $n = 6$ mice for the Senegal_2015 and F_Polynesia_2013 strains and $n = 8$ mice for the Thailand_2014 strain. Statistical significance of the differences was determined by one-way ANOVA followed by Tukey's post hoc test and is only shown when significant (***$p < 0.001$; **$p < 0.01$; *$p < 0.05$). Viral loads differed significantly between ZIKV strains ($p < 0.0277$) in all organs. Source data are provided as a Source Data file.

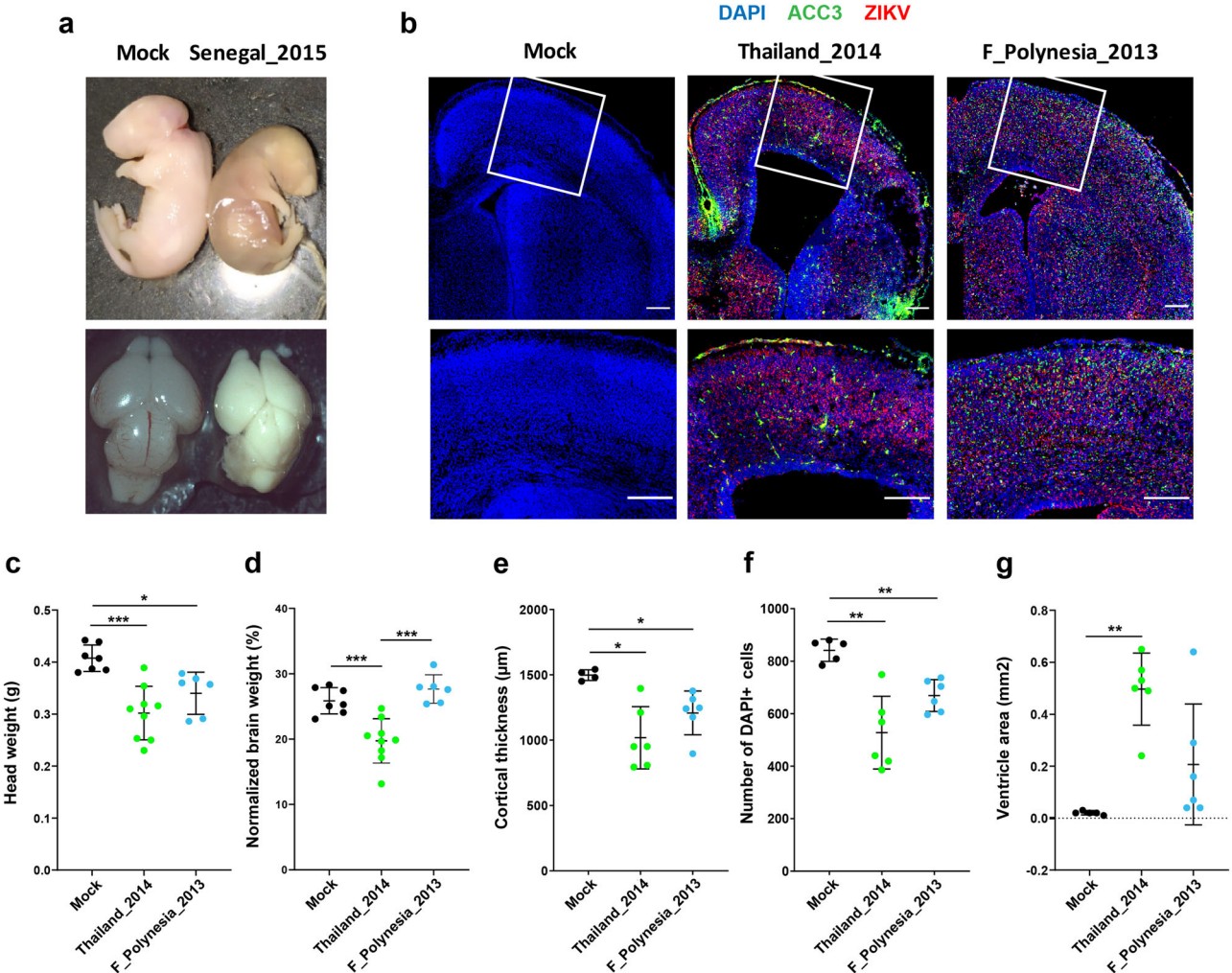

**Fig. 6 Brain phenotypes of mouse embryos vertically infected with African and Asian strains of ZIKV.** Immunocompetent mouse embryos were infected at E10.5 by intraplacental injection of 500–1000 PFU of ZIKV and analyzed at E18.5 by microdissection. **a** Representative view of E18.5 embryos (top) and dorsal view of E18.5 embryonic brains (bottom) after mock injection (left; $n = 10$) or infection by the Senegal_2015 ZIKV strain (right; $n = 7$). **b–g** Analyses of in utero brain development of E18.5 mouse embryos after mock injection ($n = 7$ for head and brain measurements and $n = 5$ otherwise) or infection by the Thailand_2014 ($n = 9$ for head and brain measurements and $n = 6$ otherwise) or the F_Polynesia_2013 ($n = 6$) ZIKV strains. **b** Immunolabeling of embryonic brain sections representative of each ZIKV strain tested (top: full view; bottom: enlarged area within white frame). Blue, green and red colors indicate DAPI, anti-cleaved caspase 3 (ACC3) and ZIKV stainings, respectively. The scale bars represent 200 μm. **c, d** Embryonic heads and brains were examined morphologically by measuring (**c**) head weight and (**d**) brain weight normalized to head weight. **e, f** Microcephalic phenotypes were assessed by measuring (**e**) cortical thickness and (**f**) number of DAPI-positive cells. **g** Ventriculomegaly was estimated by measuring the ventricle area. In (**c–g**) data are presented as mean ± standard deviation. Statistical significance of differences was determined by Brown–Forsythe and Welch ANOVA followed by Tamhane's T2 multiple comparison test (two sided). Only statistically significant differences are shown (***$p < 0.001$; **$p < 0.01$; *$p < 0.05$). Embryos infected with the F_Polynesia_2013 and Thailand_2014 ZIKV strains had a significantly smaller head weight, cortical thickness and number of cortical cells than the mock-injected embryos ($p < 0.05$). Source data are provided as a Source Data file.

mice model was not associated with the epidemic or non-epidemic status of the Asian ZIKV strains. ZIKV also has the potential to antagonize innate immune responses of the host, which may involve various ZIKV proteins[27,63–66]. The mechanism by which the antagonistic effect is brought about may be shared by ZIKV strains from both lineages[66,67], may be strain- or lineage dependent[39,68], or has only been described for specific ZIKV strains[65]. However, immunocompromised mouse models are not suitable for comparing the ability of ZIKV strains to suppress or evade the host's immune system, which may additionally contribute to their epidemic potential.

Our findings from the mosquito infection experiments and the immunocompromised mouse model support the hypothesis that worldwide ZIKV emergence in the last 15 years was not driven by adaptive virus evolution[2]. We found no evidence for enhanced transmission by the primary epidemic vector, *Ae. aegypti*, or increased level and/or duration of viremia in the vertebrate host between epidemic Asian ZIKV strains (F_Polynesia_2013 and Puerto_Rico_2015) and non-epidemic Asian ZIKV strains (Cambodia_2010, Philippines_2012 and Thailand_2014). The Asian ZIKV strain that gave rise to the widespread epidemics in the Pacific and the Americas was probably not selected for its superior ability to infect mosquitoes and/or humans. Instead, it seems more likely that it was stochastically introduced through increased air travel and human mobility in regions with favorable epidemic conditions such as high densities of competent vectors and immunologically naïve human populations[69].

The lack of human outbreaks associated with the African lineage of ZIKV until now[9] is paradoxical because a large majority of experimental studies have found a higher transmissibility and pathogenicity of the African ZIKV strains relative to their Asian counterparts[26–32]. We hypothesized that this discrepancy could have reflected the lack of recent, low-passage African strains available for experimental studies. Our panel included two ZIKV strains isolated in Senegal in 2011 and 2015, which are several decades more recent than most of the African ZIKV strains used in earlier studies. Note that our two African ZIKV strains were isolated from mosquito pools whereas the Asian strains were isolated from human serum samples, however all virus stocks were produced in mosquito cells prior to the experiments. Our study unequivocally confirms that the African lineage of ZIKV is associated with higher transmissibility and pathogenicity. The reasons why African ZIKV strains have so far not been responsible for human outbreaks (or remained unnoticed) are unknown, but may include the lack of awareness prior to the worldwide emergence, the paucity of surveillance programs in resource-poor countries, protective effects of herd immunity or cross reactive antibodies, and/or the lower vectorial capacity of African mosquito populations.

Most ZIKV infections in humans are asymptomatic and the majority of symptomatic infections cause a non-specific acute febrile illness that can easily be mistaken for other common viral infections[70]. In the absence of severe infection outcomes such as GBS or CZS, low-level ZIKV circulation or even small-scale outbreaks could have gone unnoticed, especially before the emergence in the Pacific and Americas when ZIKV was still largely unknown. The worldwide emergence of ZIKV raised international awareness, resulting in improved surveillance including the implementation of epidemiological studies in regions where the virus was known to be present prior to the pandemic. Seroprevalence studies recently conducted across Africa generally found a low level (<6.2%) of specific anti-ZIKV antibodies[71–75]. These studies are consistent with low-level ZIKV circulation in human populations of Africa and rule out the hypothesis that a high level of herd immunity is the main factor preventing ZIKV outbreaks in Africa. The alternative hypothesis that sustained circulation of closely related flaviviruses such as dengue virus (DENV) confers cross protection against ZIKV is also unlikely because high DENV seroprevalence has not prevented the emergence of ZIKV in South America, the Caribbean and the Pacific[76].

A possible explanation for the lack of ZIKV outbreaks in Africa despite the high epidemic potential of African ZIKV strains is a reduced vectorial capacity of African Ae. aegypti populations. ZIKV has been isolated from multiple mosquito species, but Ae. aegypti is considered the main vector of transmission between humans in the urban cycle[1,2]. In the absence of an efficient urban vector, human ZIKV infections in Africa would be limited to spillover transmission events from the sylvatic cycle via bridge vectors, which could explain the low level of ZIKV circulation[75,77]. Large-scale human outbreaks of dengue, yellow fever and chikungunya presumably mediated by Ae. aegypti in Africa[78,79] provide evidence that the density and human biting rate of African Ae. aegypti populations are sufficient to sustain urban transmission cycles. However, we recently discovered that African Ae. aegypti populations are significantly less susceptible to ZIKV infection than non-African populations using essentially the same panel of ZIKV strains as in the present study[80]. Although the Senegal_2015 strain was more infectious to mosquitoes relative to the other ZIKV strains, African Ae. aegypti populations were overall less susceptible than non-African Ae. aegypti populations regardless of the ZIKV strain[80]. This difference mirrors the existence of the two subspecies, Ae. aegypti aegypti and Ae. aegypti formosus, which were recognized by early taxonomists and later confirmed by modern population genetics[81]. Lower ZIKV susceptibility of the African subspecies Ae. aegypti formosus could have limited human ZIKV outbreaks in Africa in spite of the higher transmissibility of African ZIKV strains. This hypothesis is consistent with earlier reports of epidemic ZIKV transmission in Angola where the mosquito population consists of a genetic mixture of Ae. aegypti aegypti and Ae. aegypti formosus[82].

An important implication of our study is that the African lineage of ZIKV should be considered a major threat to public health. Although African ZIKV strains have so far never been associated with human ZIKV outbreaks, our results with wild-type Ae. aegypti from Colombia indicate that they may have greater epidemic potential than Asian strains if exported to regions where epidemic ZIKV transmission is realistic. We also point out that the African ZIKV strains may be associated with distinct clinical features allowing them to more easily escape surveillance systems. Our observations in an immunocompetent mouse model indicate that infection in utero by African ZIKV led to fetal death, rather than birth defects. Although this finding remains to be confirmed in humans, it is consistent with the lack of reported CZS in Africa. The only confirmed cases of birth defects in Africa were caused by ZIKV strains from the Asian lineage[83]. A few suspected cases were reported in Guinea-Bissau in 2016 where the African ZIKV lineage had been detected but these have never been confirmed[84]. On the other hand, CZS is a rare symptom and its absence may also simply reflect the lack of large-scale epidemics that would allow its detection.

Bayesian reconstruction of a dated phylogenetic tree for ZIKV[85] estimated that African and Asian lineages diverged from their common ancestor between 1814 and 1852 (95% highest posterior density interval). Although viral genetic changes are generally considered the most likely explanation for the dramatic emergence and neuroinvasiveness of ZIKV within the Asian lineage[4,85], whether divergent evolution of the African and Asian lineages was driven by differential selective pressures is still an open question[13]. The lack of a sylvatic transmission cycle of ZIKV outside Africa[2] could have played an important role in the evolutionary divergence of the two lineages. However, enzootic transmission of African ZIKV strains between sylvatic mosquito species and nonhuman primates is at odds with their higher potential for epidemic transmission by Ae. aegypti, relative to Asian ZIKV strains, which are thought to primarily alternate between Ae. aegypti and humans[2]. Possibly, introduction of ZIKV into Asia after the mid-19th century could have been accompanied by a fitness drop due to a founder effect. Identifying the nucleotide variants underlying the phenotypic differences between the African and Asian ZIKV lineages will likely prove challenging because the nucleotide divergence between the two lineages is ~12%, which translates into more than 100 different amino-acid residues across the open reading frame[32]. In contrast, for instance, the American ZIKV strains typically differ from the rest of the Asian strains by less than ~30 amino-acid residues. Moreover, the phenotypic differences between the two lineages likely result from complex combinations of genetic variants[86,87], making them less tractable by conventional methods of reverse genetics.

It also remains to be elucidated whether fetal harm has always been a possible consequence of ZIKV infection during pregnancy or whether ZIKV has recently acquired mutations conferring the ability to cause fetal harm. The recent detection of three CZS cases in Thailand and Vietnam suggest that both non-epidemic and epidemic Asian ZIKV strains are neurotropic and able to be vertically transmitted during pregnancy[20–22]. Our results, supported by recent studies performed in mouse models of vertical ZIKV

transmission[43,88], indicate that African ZIKV strains also possess the ability to cross the placenta and cause adverse perinatal outcomes. This is consistent with in vitro studies showing that ZIKV strains from both the African and the Asian lineages are capable of infecting different cell types of the placental barrier such as mid-gestation amniotic epithelial cells, cytotrophoblasts, placental villous macrophages of fetal origin (Hofbauer cells), and endothelial cells[30,89–91]. Collectively, these results reinforce the hypothesis that neurotropism and vertical transmission are not novel features of a recently emerged ZIKV variant, but rather an ancestral feature of ZIKV. In our intraplacental ZIKV challenge model, the non-epidemic Thailand_2014 strain was associated with more adverse outcomes than the epidemic F_Polynesia_2013 strain, whereas the Senegal_2015 strain led to fetal loss. Thus, ZIKV could have evolved towards attenuation by causing birth defects rather than fetal loss, supporting the counter-intuitive idea that attenuation was key to the recognition of ZIKV pathogenicity.

## Methods

### Ethics and regulatory information

*Human samples.* This study used fresh human blood to prepare mosquito artificial infectious blood meals. For that purpose, healthy blood donor recruitment was organized by the local investigator assessment using medical history, laboratory results and clinical examinations. Biological samples were supplied through the participation of healthy adult volunteers (seronegative for ZIKV) at the ICAReB biobanking platform (BB-0033-00062/ICAReB platform/Institut Pasteur, Paris/ BBMRI AO203/[BIORESOURCE]) of the Institut Pasteur in the CoSImmGen and Diagmicoll protocols, which had been approved by the French Ethical Committee Ile-de-France I. The Diagmicoll protocol was declared to the French Research Ministry under reference 343 DC 2008-68 COL 1. All human subjects provided written informed consent.

*Animal experiments.* The mouse experiments conducted at Institut Pasteur were approved by the Institut Pasteur Animal Ethics Committee (project number dap170045) and authorized by the French Ministry of Research (authorization number 12861). The Institut Pasteur animal facility had received accreditation from the French Ministry of Agriculture to perform experiments on live animals in compliance with the French and European regulations on the care and protection of laboratory animals (authorization number 75-15-01). Mouse experiments conducted in Belgium strictly followed the Belgian guidelines for animal experimentation and the guidelines of the Federation of European Laboratory Animal Science Associations. Mouse experiments were performed with the approval of the Ethical Committees of the Animal Research Center of KU Leuven (authorization number P019-2016) and of the University of Liège (authorization number 16-1837), in accordance with the guidelines of the Belgian Ministry of Agriculture, and in agreement with the European Community Laboratory Animal Care and Use Regulations (86/609/CEE, Journal Officiel des Communautés Européennes L358, 18 December 1986).

*ZIKV strains.* Seven low-passage ZIKV strains (≤5 passages in cell culture) were chosen based on their geographical origin and year of isolation to best represent the current breadth of ZIKV genetic diversity (Table S1). ZIKV strains were obtained from the World Reference Center for Emerging Viruses and Arboviruses at the University of Texas Medical Branch (PRVABC59, FSS13025), the Armed Forces Research Institute of Medical Sciences (PHL/2012/CPC-0740, THA/2014/SV0127-14), the Institut Louis Malardé in French Polynesia (PF13/251013-18), and the Institut Pasteur in Dakar (Kedougou2011, Kedougou2015). High-titered stocks were prepared and their infectious titers were measured by FFA[92,93] or by plaque assay[57] in Vero cells. For FFA, a commercial mouse anti-flavivirus group antigen monoclonal antibody (MAB10216; Merck Millipore) diluted 1:1000 in phosphate-buffered saline (PBS; Gibco Thermo Fisher Scientific) supplemented with 1% bovine serum albumin (BSA; Interchim) was used as the primary antibody. The secondary antibody was an Alexa Fluor 488-conjugated goat anti-mouse antibody (A-11029; Life Technologies) diluted 1:500 in PBS supplemented with 1% BSA.

### Genome sequencing of ZIKV strains

The consensus genome sequences of the seven ZIKV strains of the panel were obtained by high-throughput sequencing[24,92]. Briefly, RNA was extracted from virus stock using QIAamp Viral RNA Mini Kit (Qiagen) and treated with TURBO DNase (Ambion). The Senegal_2011 and Senegal_2015 strains also underwent depletion of host ribosomal RNA following a homemade protocol[24]. cDNA was produced with random hexameric primers (Roche) using M-MLV (Invitrogen) or Superscript IV (Thermo Fisher Scientific) reverse transcriptase. After second-strand synthesis with Second-Strand Synthesis Buffer (New England Biolabs), dsDNA was used for library preparation using Nextera XT DNA Kit (Illumina) or NEBNext Ultra II RNA Library Prep kit (New England Biolabs) according to the manufacturer's instructions. The final libraries were checked on a Bioanalyzer (Agilent) and combined with other libraries from unrelated projects to be sequenced on an Illumina NextSeq 500 instrument (150 cycles, paired ends). Raw sequencing datasets were deposited to the European Nucleotide Archive database under accession number PRJEB39677. The sequencing data were processed through a custom pipeline[94]. Briefly, nucleotides with a quality score <30 were trimmed using Trimmomatic v0.36 (ref. [95]). Reads were filtered against the *Aedes albopictus* reference genome using Bowtie2 v2.3.4.3 (ref. [96]) and the remaining reads were subjected to de novo assembly with the Ray v2.3.1-mpi tool[97] or metaSPAdes[98]. Scaffolds were subjected to a blastn search in the nucleotide NCBI database using BLAST v2.2.40 (ref. [99]). The closest hit was used to produce a chimeric genome sequence that served as a reference to remap the filtered reads with Bowtie2 v2.3.4.3 and generate a consensus sequence.

### Phylogenetic analyses

Genome sequences of ZIKV and Spondweni virus were retrieved from GenBank. The nucleotide sequences were aligned using MAFFT[100]. The phylogenetic analyses were performed based on nucleotide (open reading frame), amino-acid, and codon alignments using the maximum-likelihood method with substitution models (GTR + F + G4, FLU + G + R3, and SCHN05 + FU + R4, respectively) selected with ModelFinder[101]. Support for the tree was assessed with 1000 ultrafast bootstrap replicates[102]. The consensus trees were reconstructed with IQ-TREE v1.6.3 (ref. [103]) and visualized in FigTree v1.4.4 (https://github.com/ rambaut/figtree/releases). The phylogenetic tree root position was in agreement among the nucleotide-based tree without an outgroup (midpoint), amino-acid or codon tree with Spondweni virus as an outgroup, as well as with previously published ZIKV phylogenetic trees[104]. In addition to the seven genome sequences generated in this study, the alignment used to construct the tree included 37 sequences from GenBank (accession numbers: MK241416; MF574587; KX198135; KU647676; KY693679; KU497555; MF794971; KY829154; MH882540; KY014295; MH063262; KY631494; KY693677; MF434522; MF801378; KU758877; KU937936; KY693680; KU509998; KX806557; LC191864; KU963796; MF036115; LC219720; MN190155; KY241695; MH013290; MK238037; KX051562; EU545988; KX377336; MN025403; MF510857; KU963574; KF268948; MK105975; KY288905).

### Mosquitoes, mice and cell lines

*Mosquitoes.* All mosquito experiments used the 4th and 5th generations of an *Ae. aegypti* colony established from wild specimens caught in Barranquilla, Colombia, with the exception of the mouse-to-mosquito transmission experiment that used the 9th generation of an *Ae. aegypti* colony from Saint François, Guadeloupe and the 13th generation of an *Ae. aegypti* colony from La Lopé, Gabon. Mosquitoes were maintained under controlled insectary conditions (28° ± 1 °C, 12 h:12 h light: dark cycle and 70% relative humidity)[92]. Larvae were reared in dechlorinated tap water supplemented with a standard diet of Tetramin (Tetra) fish food. Adults were kept in insect cages (BugDorm) with permanent access to 10% sucrose solution.

*Mouse strains.* In-house-bred, 6- to 12-week-old male AG129 mice (Marshall BioResources, Hull, UK), deficient in both interferon (IFN)-α/β and IFN-γ receptors, were used for experimental ZIKV infections. In-house-bred, 10-week-old male and female 129S2/SvPas mice deficient for IFN-α/β receptors (*Ifnar1*$^{-/-}$), were used for the mouse-to-mosquito ZIKV transmission assay. Time-mated, wild-type immunocompetent SWISS mice (Janvier Labs, Saint Berthevin, France) of 8–12 weeks of age were used for ZIKV vertical transmission experiments. Mice were maintained under standard housing conditions (18–23 °C, 14 h:10 h light:dark cycle and 40–60% relative humidity).

*Cell lines.* The *Aedes albopictus* cell line C6/36 (ATCC CRL-1660) was used for amplification of all virus stocks and testing of mosquito saliva samples. C6/36 cells were maintained at 28 °C under atmospheric $CO_2$ in Leibovitz's L-15 medium (Gibco Thermo Fisher Scientific) with 10% fetal bovine serum (FBS), 2% tryptose phosphate broth (Gibco Thermo Fisher Scientific), 1× nonessential amino acids (Gibco Thermo Fisher Scientific), 10 U/ml of penicillin (Gibco Thermo Fisher Scientific) and 10 μg/ml of streptomycin (Gibco Thermo Fisher Scientific)[57,93]. The *Cercopithecus aethiops* cell line Vero (ATCC CCL-81) was used for titration of virus stocks by FFA. The *C. aethiops* cell line Vero E6 (ATCC CRL-1586) was used for titration of virus stocks by plaque assay. Vero cells were maintained at 37 °C under 5% $CO_2$ in Dulbecco's Modified Eagle Medium (Gibco Thermo Fisher Scientific) with 10% FBS, 10 U/ml of penicillin, and 10 μg/ml of streptomycin[57,93].

### Mosquito exposure to ZIKV via artificial blood meals

Mosquitoes were orally challenged with ZIKV by membrane feeding[92]. Briefly, 7-day-old females deprived of sucrose solution for 24 h were offered an artificial infectious blood meal for 15 min using a Hemotek membrane-feeding apparatus (Hemotek Ltd.) with porcine intestine as the membrane. Blood meals consisted of a 2:1 mix of washed human erythrocytes and ZIKV suspension. Adenosine triphosphate (Merck) was added to the blood meal as a phagostimulant at a final concentration of 10 mM. Fully engorged females were sorted on wet ice, transferred into 1-pint cardboard containers and maintained under controlled conditions (28° ± 1 °C, 12 h:12 h light:dark cycle and 70% relative humidity) in a climatic chamber with permanent access to

10% sucrose solution. After 7, 10, 14, and 17 days of incubation, mosquitoes were paralyzed with triethylamine to collect their saliva in vitro. The proboscis of each female was inserted into a 20 µl pipet tip containing 10 µl of FBS. After 30 min of salivation, the saliva-containing FBS was mixed with 30 µl of Leibovitz's L-15 medium (Gibco Thermo Fisher Scientific), and stored at −80 °C for later testing. The saliva samples were subsequently thawed and inoculated onto C6/36 cells for ZIKV detection by FFA as described above without subsequent dilution. Mosquito bodies were homogenized individually in 300 µl of squash buffer (Tris 10 mM, NaCl 50 mM, EDTA 1.27 mM with a final pH adjusted to 8.2) supplemented with 1 µl of proteinase K (Eurobio Scientific) for 55.5 µl of squash buffer. The body homogenates were clarified by centrifugation and 100 µl of each supernatant were incubated for 5 min at 56 °C followed by 10 min at 98 °C to extract viral RNA. Detection of ZIKV RNA was performed using a two-step RT-PCR reaction to generate a 191-bp amplicon located in a conserved region of the ZIKV genome between the 3′ end of the NS1 gene and the 5′ end of the NS2A gene. Total RNA was reverse transcribed into cDNA using random hexameric primers and the M-MLV reverse transcriptase (Thermo Fisher Scientific) according to the following program: 10 min at 25 °C, 50 min at 37 °C, and 15 min at 70 °C. The cDNA was subsequently amplified using DreamTaq DNA polymerase (Thermo Fisher Scientific). For this step, 20 µl reaction volumes contained 1× of reaction mix and 10 µM of each primer, whose sequences are provided in Table S3. The thermocycling program consisted of 2 min at 95 °C, 35 cycles of 30 s at 95 °C, 30 s at 60 °C, and 30 s at 72 °C with a final extension step of 7 min at 72 °C. Amplicons were visualized by electrophoresis on a 2% agarose gel.

**Mouse-to-mosquito ZIKV transmission assay.** Ten-week-old 129S2/SvPas $Ifnar1^{-/-}$ mice (males and females) were intraperitoneally injected with a 200 µl inoculum containing $10^5$ FFU of ZIKV (Cambodia_2010 strain). From day 1 to day 5 post inoculation, mice were anesthetized daily using 80 mg/kg of ketamine and 5 mg/kg of xylazine administered by the intraperitoneal route. Each anesthetized mouse was placed on the netting-covered top of two 1-pint cardboard boxes, each containing 25 2- to 4-day-old Ae. aegypti females either from Guadeloupe or Gabon colonies, which differ significantly in their ZIKV susceptibility[80]. Mosquitoes previously deprived of sucrose solution for 24 h were allowed to blood feed on the mouse for 15 min. Fully engorged females were sorted on wet ice, transferred into fresh 1-pint cardboard containers and maintained under controlled conditions (28° ± 1 °C, 12 h:12 h light:dark light cycle and 70% relative humidity) in a climatic chamber with permanent access to 10% sucrose solution. After 14 days of incubation, saliva and bodies were collected and analyzed by FFA and RT-PCR as described above.

**ZIKV experimental infections of immunocompromised mice**
*Survival.* AG129 mice (6- to 12-week-old males) were intraperitoneally injected with $10^3$ PFU of ZIKV (2 × 5 mice per ZIKV strain). Upon infection, mice were observed daily for changes in body weight and clinical symptoms of virus-induced disease including dehydration, hunched back, and paralysis. Mice were euthanized when body weight loss was >20% or when other humane endpoints were met according to the ethical guidelines. During days 1–8 of infection, mice were bled by submandibular puncture to monitor viremia kinetics. Plasma viremia was measured every other day from two alternating subgroups of five mice each. Upon euthanasia, the brain, spinal cord, testis, and epididymis were collected and blood was collected by intracardiac puncture. After collection, tissues were immediately placed on dry ice and stored at −80 °C until further processing.

*Tissue tropism.* AG129 mice (6- to 12-week-old males) were intraperitoneally injected with 1 PFU (8 mice per ZIKV strain). Upon infection, mice were observed daily for changes in body weight and clinical symptoms of virus-induced disease in case euthanasia would be required based on humane endpoints. Mice were bled by submandibular puncture on days 3, 4, and 5 post infection. On day 7 post infection, all animals were sacrificed and blood was collected by intracardiac puncture. All animals were euthanized by intraperitoneal injection of Dolethal (Vétoquinol). The brain, spinal cord, testis, epididymis, heart, liver and kidney were collected after transcardial perfusion with PBS. After collection, tissues were immediately placed on dry ice and stored at −80 °C until further processing.

**ZIKV vertical transmission in immunocompetent mice**
*Intraplacental injections.* Timed-mated, wild-type pregnant SWISS dams (8- to 12-week-old) were housed under standard conditions and allowed to acclimate for at least 24 h upon receipt with access to food and water ad libitum. The surgeries were performed at the same time of day, with noon of the day after mating set as embryonic (E) 0.5. Preoperative analgesia was administered subcutaneously with 0.1 mg/kg of buprenorphine (Temgesic) before induction of anesthesia with isoflurane (Abbot Laboratories Ltd.) in an oxygen carrier. A 1.0- to 1.5 cm incision was performed through the lower ventral peritoneum and the uterine horns were careful extracted onto warm humidified gauze pads. The intraplacental injections of embryos were performed at E10.5 (ref. [58]). The fast green dye concentration was 0.05% and placenta was injected with either ZIKV or mock medium. The animals were randomly assigned to receive a 1.0 to 2.0 µl injection of mock medium or ZIKV stock containing $5 × 10^5$ PFU/ml.

*Immunohistochemistry.* After dissection, E18.5 mouse heads and E14.5 embryos were fixed in 4% paraformaldehyde in PBS for 24 h at 4 °C. Brains were dissected in 0.1 M PBS (pH 7.4). E18.5 brains and E14.5 embryos were cryoprotected (20% sucrose in PBS) before being embedded in OCT (Richard-Allan Scientific Neg-50 Frozen Section Medium, Thermo Scientific) for cryosectioning 14 µm sections for brains and 20 µm sections for embryos (Leica) onto slides (SuperFrost Plus, VWR International). For fluorescence immunohistochemistry[58], a solution of antigen retrieval (Dako Target Retrieval Solution) was pre-heated at 95 °C for 40 min and antigen retrieval of mouse brains and whole embryos were performed at 95 °C for 5 min before incubation with primary antibodies. The primary antibodies were rabbit anti-cleaved caspase 3 (1:300, #9661, Cell Signaling Technologies), mouse anti-flavivirus group antigen (1:800, MAB10216, Merck Millipore) and goat anti-Iba1 (1:300, ab5076, Abcam). The respective secondary antibodies were donkey anti-rabbit, anti-mouse and anti-goat antibodies conjugated with Alexa Fluor-488, Alexa Fluor-555, and Alexa Fluor-647 (A-21206, A-31570, A-21447, Life Technologies) and diluted 1:1000. Nuclei were counterstained with DAPI (1:1000, Sigma) and mounted in Dako Fluorescence Mounting Medium (Agilent).

*Image acquisition and processing.* Immunofluorescence images of embryonic brains (E18.5) and internal organs (E14.5) were acquired in magnified fields (×20 and ×25) with either Nikon A1 or Zeiss LSM 880 AiryScan Elyra S.1 confocal microscopes and further processed with ImageJ 1.42q 276 (Wayne Rasband, National Institutes of Health), Fiji (v2.0.0-rc-54/1.51 h, https://imagej.net/Fiji) and Zen (Blue edition, Carl Zeiss Microscopy GmbH) software.

**ZIKV detection and quantification in mouse samples**
*Quantification of ZIKV RNA by qRT-PCR.* Total RNA was isolated from microdissected embryonic mouse (E14.5) tissues using Trizol (Ambion, Life Technologies) according to the manufacturer's protocol. For adult mouse samples, sections of whole tissue were weighed and transferred to 2 ml Precellys tubes containing 2.8 mm zirconium oxide beads (Bertin Instruments). RLT lysis buffer (RNeasy Mini Kit, Qiagen) was added at a ratio of 19 times the weight of the tissue section. Tissue sections were homogenized in three cycles at 6800 rpm with 30 s intervals using the Precellys24 homogenizer (Bertin Instruments). Homogenates were cleared by centrifugation (10 min, 14,000 rcf, 4 °C) and total RNA was extracted from the supernatant using the RNeasy Mini Kit (Qiagen) according to the manufacturer's protocol. For plasma samples, viral RNA was extracted using the NucleoSpin RNA kit (Macherey-Nagel) following the manufacturer's instructions. Viral RNA was eluted in 50 µl of RNase-free water. Quantification of ZIKV genome copy numbers was performed by quantitative reverse transcription PCR (qRT-PCR) using the Applied Biosystems 7500 Fast Real-Time PCR System (Thermo Fisher Scientific). The Asian ZIKV strains were detected and quantified using a specific primer pair and a double-quenched probe (Integrated DNA Technologies). The African ZIKV strains were detected and quantified using the same probe as for the Asian ZIKV strains but a different primer pair to accommodate several mismatches. Standard curves were generated based on tenfold serial dilutions of gBlock synthetic oligonucleotides (Integrated DNA Technologies) whose sequences were specific to the Asian and African ZIKV lineages. Ct values were converted into a relative number of ZIKV RNA copies/mg of tissue or ml of plasma using the formula $y = a*\ln(x) + b$, where $a$ is the slope of the standard curve, $b$ is the $y$-intercept of the standard curve and $y$ is the Ct value for a specific sample.

*Endpoint titration of infectious ZIKV.* The amount of infectious virus present in brain and testis samples was estimated by 50% tissue-culture infectious dose $(TCID_{50})$ endpoint titration. Tissue sections were weighed and homogenized in 500 µl of Vero E6 cell culture medium (with 2% FBS) as described above. The homogenates were cleared by centrifugation (10 min, 14,000 rcf, 4 °C). Vero E6 cells were seeded at a density of $1 × 10^4$ cells/well in 96-well microtiter plates and incubated overnight. The next day, they were inoculated with triplicate tenfold serial dilutions of the supernatant samples. After 7 days of incubation, the cells were examined microscopically for virus-induced cytopathic effects. A well was scored positive if any signs of virus-induced cytopathic effects were observed compared to the uninfected control cells. The $TCID_{50}$/mg tissue was calculated using the method of Reed and Muench[105].

**Statistics.** Statistical analyses were performed using JMP v10.0.2 (www.jmpdiscovery.com), GraphPad Prism v8.02 (www.graphpad.com) and the packages *car*, *userfriendlyscience* and *DescTools* of R v3.6.0 (www.r-project.org). Binary variables were analyzed by logistic regression followed likelihood-ratio $\chi^2$ tests. Body weight and $\log_{10}$-transformed viremia levels were compared by repeated measures analysis (restricted maximum-likelihood method) using a mixed model in which ZIKV strain was nested within lineage, and mouse (random effect) was nested within ZIKV strain. Other continuous variables were analyzed by analysis of variance (ANOVA) when the underlying assumptions were satisfied. They were analyzed by Brown–Forsythe and Welch ANOVA when the homoscedasticity assumption was unmet and by Kruskal–Wallis rank-sum test when both the normality and the homoscedasticity assumptions were unmet. Survival curves were

compared by log-rank test. Differences were considered statistically significant when $p < 0.05$.

### Epidemiological modeling

*Model overview.* To assess the epidemiological potential of ZIKV strains from empirically observed variation in mosquito infection dynamics, stochastic agent-based simulations were performed using the R package *nosoi*[54] (available from https://github.com/slequime/nosoi). This modeling framework accounts for the influence of within-host infection dynamics on transmission probability during mosquito–human infectious contacts in a full epidemiological context[106]. Virus transmission was assumed to only occur either between an infected human and an uninfected mosquito, or between an infected mosquito and an uninfected human. Sexual and vertical transmission modes were considered epidemiologically insignificant[107,108] and ignored in the model, and so were super-infections. The human and mosquito populations were considered homogeneous. Mosquito daily survival probability was set to 0.85 (ref. [109]). The daily number of mosquitoes biting a human was drawn from a Poisson distribution ($\lambda = 2.1$) assuming a daily biting probability of 0.7 (ref. [110]) and a relative mosquito density of 3 (ref.[109]).

*Human-to-mosquito transmission.* Probabilities of human-to-mosquito ZIKV transmission were inferred from empirical data obtained with the mouse-to-mosquito transmission assay described above (Fig. S1). Mean transmission probabilities ($\mu$) and their standard deviations (sd) were estimated from the infection rates of mosquitoes tested by RT-PCR 14 days post infectious blood meal as described above. Transmission probabilities were used to derive a beta distribution of parameters $\alpha = ((1 - \mu)/sd^2 - 1/\mu)*\mu^2$ and $\beta = \alpha*(1/\mu - 1)$ from which daily transmission probability for each simulated human was drawn until 6 days post infection. The parameters governing human-to-mosquito transmission were shared among all ZIKV strains.

*Mosquito-to-human transmission.* The cumulative proportion of infectious mosquitoes over time for each ZIKV strain was described by a two-parameter log-logistic model using the *drm* function in the R package *drc*[111]. This model (LL.2 equation in the *drm* function) has a lower limit fixed at 0 and an upper limit fixed at 1 and is given by the formula: $f(x, (b, e)) = 1/(1 + \exp(b * \log(x) - e))$. The probability of mosquito-to-human transmission for each contact between a human and an infected mosquito was determined by the predicted transmission probability inferred for each ZIKV strain. For each mosquito, an individual extrinsic incubation period (EIP) was drawn from the empirically determined log-logistic distribution of EIP values (location $= \log(e)$, scale $= 1/b$). The EIP value was used as a threshold for the individual probability of virus transmission over time. Transmission probability was 0% before the EIP and 100% after the EIP.

*Implementation.* Epidemiological simulations were run in R v3.6.0 using the packages *foreach* (https://cran.r-project.org/web/packages/foreach/foreach.pdf), *doParallel* (https://cran.r-project.org/web/packages/doParallel/doParallel.pdf), and *nosoi*[54]. The custom code is provided in Supplementary Software 1. Briefly, simulations were initiated with one infected human and zero infected mosquitoes and run for 400 days or until 100,000 humans or 1,000,000 mosquitoes were reached, whichever occurred first. A total of 100 independent replicate simulations were run for each ZIKV strain.

**Reporting summary**. Further information on research design is available in the Nature Research Reporting Summary linked to this article.

## Data availability

Raw sequence data generated in this study were deposited in the European Nucleotide Archive database under accession number PRJEB39677 (https://www.ebi.ac.uk/ena/browser/view/PRJEB39677). Phylogenetic analysis of virus genomes used GenBank accession numbers MK241416, MF574587, KX198135, KU647676, KY693679, KU497555, MF794971, MK829154, MH882540, KY014295, MH063262, KY631494, KY693677, MF434522, MF801378, KU758877, KU937936, KY693680, KU509998, KX806557, LC191864, KU963796, MF036115, LC219720, MN190155, KY241695, MH013290, MK238037, KX051562, EU545988, KX377336, MN025403, MF510857, KU963574, KF268948, MK105975, and KY288905. Source data are provided with this paper.

## Code availability

The custom code is provided in Supplementary Software 1. The R package *nosoi* is available from https://github.com/slequime/nosoi.

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

## Acknowledgements
We thank Catherine Lallemand for assistance with mosquito rearing, the Institut Pasteur animal facility staff for the breeding of *Ifnar1*$^{-/-}$ mice. We thank Diego Ayala, Christophe Paupy, and Davy Jiolle for initially providing the mosquito colony from Gabon and Anubis Vega-Rúa for initially providing the mosquito colony from Guadeloupe. We are grateful to the volunteers and the ICAReB staff for the human blood supply. We thank Elke Maas, Carolien De Keyzer, and Lindsey Bervoets for technical assistance with the AG129 mouse experiments, the KU Leuven Rega animal facility staff for breeding the AG129 mice, and Jelle Matthijnssens and Daan Jansen for resequencing the ZIKV strains at the KU Leuven Rega Institute. We thank Alexandre Hego and the GIGA-Imaging platform for their precious help and advice, and Christian Alfano and Ivan Gladwyn-Ng for their guidance with the vertical transmission experiments. This work was primarily funded by the European Union's Horizon 2020 research and innovation program under ZikaPLAN grant agreement no. 734584 (to L.L. and J.N.) and under ZIKAlliance grant agreement no. 734548 (to L.N. and J.N.). This work was also supported by Agence Nationale de la Recherche (grants ANR-16-CE35-0004-01, ANR-17-ERC2-0016-01, and ANR-18-CE35-0003-01 to L.L.), the French Government's Investissement d'Avenir program Laboratoire d'Excellence Integrative Biology of Emerging Infectious Diseases (grant ANR-10-LABX-62-IBEID to L.L. and X.M.), the Inception program (Investissement d'Avenir grant ANR-16-CONV-0005 to L.L.), the Research Foundation Flanders (FWO Ph.D. fellowship 1S21918N to S.J.), and the Fonds de la Recherche Scientifique (FRIA Ph.D. fellowship to M.D.). The funders had no role in study design, data collection and interpretation, or the decision to submit the work for publication.

## Author contributions
S.J.F.K. and L.L. contributed equally to the design and coordination of the study. F.A., N.J., E.F.M., S.D., C.M., and A.B. carried out the mosquito experiments. S.L and A.F. conducted the epidemiological modeling. S.J., L.D., and S.J.F.K. designed and performed the AG129 mouse experiments. M.D. and M.G. designed and performed the vertical transmission experiments. S.D., S.L., F.G., and E.S.-L. performed the virus sequencing. F.A. and A.B. performed the phylogenetic analyses. C.M.R.-V. conducted the field collections to initiate the mosquito colony from Colombia. V.-M.C.-L., R.G.J., C.T.D., Oum. F., Ous.F., and A.A.S. obtained the virus isolates and organized their transfer. F.A., C.M., X.M., and L.L. designed and implemented the mouse-to-mosquito transmission assay. J.N. supervised the AG129 mouse experiments. L.N. supervised the vertical transmission experiments. F.A., S.J., M.D., S.J.F.K., and L.L. prepared the figures. F.A. and L.L. analyzed the data and wrote the paper with input from all other authors.

## Competing interests
The authors declare no competing interests.
