## [Peer Review File · Nature Communications]

Reviewers' Comments:

Reviewer #1:

Remarks to the Author:

Review of manuscript NCOMMS-20-33056-T.

“High transmissibility and fetal pathogenicity of recent Zika virus strains from the African lineage”

Comments to the editor and authors: Zika virus (ZIKV) is a vector-borne flavivirus that is emerging as an important human pathogen of global scale and causing outbreaks associated with severe and often fatal disease in humans. Despite a progress in recent studies of ZIKV epidemiology, there is completely unclear phenomenon: why all human ZIKV have been entirely connected with the Asian lineage of ZIKV strains? The manuscript by Aubry et al. based on reasonably well documented research data on the difference between African and Asian ZIKV strains in mosquito transmission and virulence in mouse animal models. Extensive analysis of low-passaged ZIKV viruses in mosquitoes and mice allows authors to differentially distinguish African ZIKV from Asian strains in terms of their significantly higher transmissibility in *Aedes aegypti*, the severity of induced pathology in immunocompromised mice, and magnitude of virus impact on embryonic CNS development. The findings in this paper are important in improving the understanding of ZIKV pathogenesis and provide a basis to consider the African lineage of ZIKV as a potential human pathogen source for new epidemics and as well as a serious threat for public health. The work has been very neatly done, and the presentation and conclusions are well supported by the data. I have no major criticisms or concerns and support this manuscript for publication in Nature Communication with minor revisions.

Few points or general considerations listed below are of a minor nature and can be addressed in the paper to improve clarity of the statements.

1) Authors, please discuss general limitations of mouse models for ZIKV infection and limitation of immunodeficient models such as AG129 mice for studies of ZIKV pathogenesis. I am not convinced that specific differences in various aspects of pathogenicity among African and Asian ZIKV strains, that were observed in mice, can be directly translated to humans. Would a non-human primate (NHP) model of ZIKV infection be more relevant for such studies? In the 'Introduction', author cited three studies using NHP (refs 49-51) which shows that some Asian strains have better fitness in NHP as compared to African ZIKV. What might be possible explanation for these observations?

2) In the 'Results' and/or 'Discussion' section, please discuss limitations of inoculating virus to the labyrinth of mouse placenta. In my understanding, labyrinth is an embryonal part of mouse placenta, and this suggests that injected virus does not need to traverse a placental barrier and can freely infect all organs of developing fetus with a blood flow. However, ability to traverse/cross a placental barrier might be the key difference determining teratogenicity African and Asian strains. For instance, if African strains are not very efficient in the traversing of placenta barrier, then they could get a chance to cause developmental defects.

3) Both Asian and African strains are descendants of common ancestor. It appears that since the time of lineage separations either 1) African strains gained fitness advantages or 2) Asian ZIKV strains lost fitness in both vertebrate and mosquito host (which I think is more likely scenario). In the any case, could authors provide a potential explanation on to why ZIKV was subjected to a different evolutionary pressure in Asia and Africa. In addition, can authors provide a citation any studies on ZIKV phylogenetic, which estimates an approximate time of separation of ZIKV lineages.

Reviewer #2:

Remarks to the Author:

The authors characterize two recent isolates of Zika virus from Senegal. The authors show increased mosquito infection and potential transmission rates as well as higher virulence in a murine model of the two isolates than recent Asian/American isolates.

The results of the study are not surprising as Zika is endemic in parts of West Africa. Previously, multiple groups have performed mosquito and animal studies with an isolate of Zika DAK41525 isolated in 1984 in Senegal, and have shown higher mosquito infection rates, virulence in multiple murine models, fetal abnormalities, and sexual transmission rates in three non-human primate species (African green, rhesus, and cynomolgus). Consequently, these results are in agreement with the previous findings and suggest lineage specific differences.

The present study has several issues that need to be addressed with additional experiments:

1) Infectious virus need to be quantitated by plaque assay in both mosquito and murine studies. This is considerable limitation in the study.

a) Infection rates in mosquitoes should be determined by infectious virus assays as the presence of RNA dose not correspond to productive infection.

b) The presence of RNA in the saliva does not correspond to transmission of infectious virus.

c) In both figs. 4D and 5B, infectious virus needs to be quantitated.

2) The concentration of infectious virus concentration present immediately after oral infection to determine the ingested dose.

3) Earlier time points prior to 7 dpi for the mosquito studies need to be performed. In Fig. 4A, all isolates show a high rate of RNA in the mosquitoes.

4) The use of Asian isolates in the high and low dose mosquito study is puzzling. Five isolates are used for high dose, and the low dose utilizes only one isolate. The low dose mosquito experiment should be performed with other Asian isolates.

5) The murine studies in Fig. 4 show that Thailand and French Polynesian are the two most attenuated isolates relative to the Senegal isolates. The subsequent experiments in Fig 4C and D to determine tissue viral load and mouse embryo experiments in Fig 5 will predictably yield greater differences than the African isolates. This is a puzzling choice of Asian isolates for Figs 4 and 5. Similar is true for Fig. 6.

Reviewer #3:

Remarks to the Author:

The manuscript reports a study on Zika virus transmission using experimental transmission trials as well as epidemiological modelling. The objectives were (i) to compare the transmissibility of Asian and African lineage, (ii) to assess the ability to disrupt embryonic development. This study covers a wide range of aspects, which allow for a better characterization of the ZIKV transmission processes at host and vector scales and pathogenicity. Finally, the authors successfully combined multiple disciplines in a comprehensive way. Although my field of expertise is related with mathematical modelling, I appreciated the whole manuscript and the global approaches the authors used. The modelling section is also well described with the code and ad-hoc references. I only have minor comments:

- The study aims to compare African and Asian lineage in terms of transmissibility and pathogenicity. The title should reflect this comparison.

- Some parts of the result section would better stand (and may be somehow redundant with) Material and methods. For example, lines 157-163, or 215-231.

- Statistical analyses (L 262-284): Data are compared for each time point. Here we have longitudinal data with non-independent samples on each individual. A global analysis on the whole trajectory using mixed effect model would allow to compare more rigorously the results. Maybe a survival analysis could have been used.

- Line 296 . Why "since"? I do not see the causality here.

- L484-497. The authors explain here that the absence of large outbreaks in Africa are likely due to

lower susceptibility of *A. Aegypti Formosus* mosquitoes. Reference 71 (which has to be updated in the reference list) supports this fact, providing some quantitative data. Could these data be used to feed the model to assess if this assumption is sufficient to explain the absence of large outbreaks in Africa? - Line 828. The authors mentioned a non-linear regression to estimate parameters K, B and M. Some words should be given in the result section and a table for the parameters could be given.

Reviewer #4:

Remarks to the Author:

In this manuscript Aurby F and co-authors performed in-depth analysis of the viral fitness of Asian and African strains of ZIKV in mosquitoes and mouse model in order to understand whether the ability of the epidemic properties of the Asian strains are determined by their adaptive evolution resulting in higher virulence or better transmission. Authors found that despite much higher rate of outbreaks, Asian strains of ZIKV are less virulent and pathogenic in mice and have lower transmission capacity compared to the African strains. Authors therefore link outbreaks of Asian ZIKV to the socio-geographical factors. Even though it has been well-established that African strains have higher viral fitness than Asian strains, all previous studies used lab-adapted old African strains that were passaged multiple times in cell lines and animals, which could have increased their virulence in the laboratory models. The study of Aurby et al used recent isolates of ZIKV (Senegal 2011, 2015) that had undergone only 2 passages in C6/36 cells, resulting for the first time in thorough characterization of African ZIKV lineage which is not confounded by the lab adaptation. This brings novelty and significance to this study even though its conclusions are not different to the current knowledge. This is a well-executed study and well-prepared manuscript with knowledgeable use of statistical and computational methods and clear data presentation, which made it a pleasure to read. However, I have a number of concerns that need to be addressed:

1. All Asian strains used in this study had higher passage number (at least 4 passages) as well as in different cells/hosts than both African strains which had only 2 passages in C6/36 cells. Therefore, the emphasis in the manuscript on low passage history for all strains needs to be toned down. The other difference between Asian and African strains is the source of the virus – all Asian strains were isolated from human serum, while both African strains were isolated from mosquitoes. This could have significant influence on viral properties. Both points need to be made clear in the text and also discussed.

2. As a model of viral pathogenesis in mammalian host, authors used AG129 mice that are highly immunocompromised and embryos of WT mice, which have not yet fully developed the immune system. Therefore, if epidemic properties of the Asian viral strains are determined by better immune suppression or evasion, this cannot be elucidated in the utilised models. I appreciate that fully immunocompetent animal models of ZIKV infection are not available, but believe that authors should discuss this limitation in the manuscript.

3. Given that all strains were fully sequenced in this study I was surprised not to see any mentioning of which changes may be related to the differences in viral properties. Surely this would be of significant interest to the authors themselves and also to the research community. At least a Table presenting key changes between the strains and some discussion on potential role of them needs to be included. I would also expect to see some data on how at least the key differences in the sequence influence viral properties. Infectious clones are now available for various ZIKV strains and could be used to answer these questions.

Reviewer #1 (Remarks to the Author):

Comments to the editor and authors: Zika virus (ZIKV) is a vector-borne flavivirus that is emerging as an important human pathogen of global scale and causing outbreaks associated with severe and often fatal disease in humans. Despite a progress in recent studies of ZIKV epidemiology, there is completely unclear phenomenon: why all human ZIKV have been entirely connected with the Asian lineage of ZIKV strains? The manuscript by Aubry et al. based on reasonably well documented research data on the difference between African and Asian ZIKV strains in mosquito transmission and virulence in mouse animal models. Extensive analysis of low-passaged ZIKV viruses in mosquitoes and mice allows authors to differentially distinguish African ZIKV from Asian strains in terms of their significantly higher transmissibility in *Aedes aegypti*, the severity of induced pathology in immunocompromised mice, and magnitude of virus impact on embryonic CNS development. The findings in this paper are important in improving the understanding of ZIKV pathogenesis and provide a basis to consider the African lineage of ZIKV as a potential human pathogen source for new epidemics and as well as a serious threat for public health. The work has been very neatly done, and the presentation and conclusions are well supported by the data. I have no major criticisms or concerns and support this manuscript for publication in Nature Communication with minor revisions.

Few points or general considerations listed below are of a minor nature and can be addressed in the paper to improve clarity of the statements.

1) Authors, please discuss general limitations of mouse models for ZIKV infection and limitation of immunodeficient models such as AG129 mice for studies of ZIKV pathogenesis. I am not convinced that specific differences in various aspects of pathogenicity among African and Asian ZIKV strains, that were observed in mice, can be directly translated to humans. Would a non-human primate (NHP) model of ZIKV infection be more relevant for such studies? In the 'Introduction', author cited three studies using NHP (refs 49-51) which shows that some Asian strains have better fitness in NHP as compared to African ZIKV. What might be possible explanation for these observations?

Response: Although NHP models may best recapitulate ZIKV pathogenesis in humans because they are genetically more related to humans, they also come with major drawbacks, including high costs and ethical constraints. This generally leads to small group sizes and thus to studies that are underpowered to detect potentially meaningful biological effects. We acknowledge that the immunocompromised AG129 mouse model may not represent the best model to study the pathogenesis of ZIKV disease. However, because AG129 mice are very susceptible to ZIKV (as well as other flaviviruses), their susceptibility can be leveraged to study the intrinsic potential of ZIKV strains to cause viremia and/or disease. It is important to note that in this case the AG129 mice are not saturated by an excessive virus inoculum. We first performed pilot studies (not included in the manuscript) to determine the optimal inoculum to (i) enable discrimination between different ZIKV strains, and (ii) allow viral replication to detectable levels. The optimal inoculum was found to be 1 PFU (lines 283-287).

To elaborate further on the pros and cons of immunodeficient mouse models, we added the following paragraph to the discussion section (lines 428-436): “Assessing ZIKV pathogenicity in the vertebrate host is complicated by the limited number of animal models that are available. Non-human primate infections closely emulate human infections but they raise ethical issues and are generally restricted to vaccine and drug development (Osuna CE & Whitney JB J Infect Dis 2017). Several models of ZIKV pathogenesis in adult mice have been developed that recapitulate various features of human disease (Lazear HM et al. Cell Host Microbe 2016; Aliota MT et al. PLoS NTDs 2016; Rossi SL et al. Cell Stem Cell 2016). In general, wild-type mice can be infected with ZIKV but they do not develop overt clinical illness and little or no viremia (Lazear HM et al. Cell Host Microbe 2016). In contrast, mice lacking the ability to produce or respond to type I interferon typically develop severe neurological disease associated with high viral loads in key organs and substantial lethality.”

The three cited NHP studies showing that some Asian ZIKV strains can display better fitness than African strains used either the IbH_30656 ZIKV strain isolated in Nigeria in 1968 (refs. 50-51) or the prototype MR766 strain (ref. 49), which has been passaged >100 times in suckling mouse brains following its isolation in Uganda in 1947. Although we cannot rule out an effect of the animal model itself, the differences observed between these studies and our study could reflect the genetic divergence and/or the age difference of the strains. We mentioned this explanation in the introduction section on lines 115-118.

2) In the ‘Results’ and/or ‘Discussion’ section, please discuss limitations of inoculating virus to the labyrinth of mouse placenta. In my understanding, labyrinth is an embryonal part of mouse placenta, and this suggests that injected virus does not need to traverse a placental barrier and can freely infect all organs of developing fetus with a blood flow. However, ability to travers/cross a placental barrier might be the key difference determining teratogenicity African and Asian strains. For instance, if African strains are not very efficient in the traversing of placenta barrier, then they could get a chance to cause developmental defects.

Response: The Reviewer is correct, the placental labyrinth is a structure on the fetal side of the placenta. Thus, we agree that our model of ZIKV vertical transmission alone does not rule out the possibility that the African ZIKV strains are physiologically less able to cross the placental barrier in humans than the Asian ZIKV strains. However, the notion that ZIKV strains from both lineages are capable of crossing the placental barrier is supported by several other *in vitro* studies on the ZIKV susceptibility of the different layers of the placental barrier. For instance, ZIKV strains from both the African and Asian lineages were able to infect different cell types of the placental barrier such as midgestation amniotic epithelial cells, cytotrophoblasts, placental macrophages (Hofbauer cells) and endothelial cells (Sheridan MA et al. PNAS 2017; Sheridan MA et al. PLoS One 2018; Tabata T et al. Cell Host Microbe 2016; Tabata T et al. J Infect Dis 2017). In addition, a recent study showed that an African ZIKV strain (IbH_30656, Nigeria 1968) was able to infect mouse fetuses when injected into the myometrium, which is the maternal part of the placenta (Vermillion MS et al. Nat Commun 2017). Finally, another recent study showed that ZIKV strains from both the Asian and African lineages could be vertically transmitted and cause fetal harm in a mouse

model (Jaeger AS et al. PLoS Negl Trop Dis 2019). Collectively, these results indicate that ZIKV lineages do not display a major divergence in their intrinsic ability to cross the placental barrier.

To address this important point, we expanded this part of the discussion and included references to the *in vitro* studies (lines 570-575), in addition to the references to the *in vivo* studies that were already mentioned (lines 568-570).

3) Both Asian and African strains are descendants of common ancestor. It appears that since the time of lineage separations either 1) African strains gained fitness advantages or 2) Asian ZIKV strains lost fitness in both vertebrate and mosquito host (which I think is more likely scenario). In the any case, could authors provide a potential explanation on to why ZIKV was subjected to a different evolutionary pressure in Asia and Africa. In addition, can authors provide a citation any studies on ZIKV phylogenetic, which estimates an approximate time of separation of ZIKV lineages.

Response: Bayesian reconstruction of a dated phylogenetic tree for ZIKV (Pettersson JHO et al. mBio 2016) estimated that African and Asian lineages diverged from their common ancestor between 1814 and 1852 (95% highest posterior density interval). Although viral genetic changes are generally considered the most likely explanation for the dramatic emergence and neuroinvasiveness of ZIKV within the Asian lineage (Pettersson JHO et al. mBio 2016; Musso D & Gubler DJ Clin Microbiol Rev 2016), whether divergent evolution of the African and Asian lineages was driven by differential selective pressures is still an open question (Liu ZY et al. Nat Microbiol 2019). The lack of a sylvatic transmission cycle of ZIKV outside Africa (Gutiérrez-Bugallo G et al. Nat Ecol Evol 2019) could have played an important role in the evolutionary divergence of the two lineages. However, enzootic transmission of African ZIKV strains between sylvatic mosquito species and non-human primates is at odds with a higher potential for epidemic transmission by *Ae. aegypti*, relative to Asian ZIKV strains, which are thought to primarily alternate between *Ae. aegypti* and humans (Gutiérrez-Bugallo G et al. Nat Ecol Evol 2019). Possibly, introduction of ZIKV into Asia after the mid-19th century could have been accompanied by a fitness drop due to a founder effect.

To address this point, we added the above paragraph to the discussion section on lines 540-553.

Reviewer #2 (Remarks to the Author):

The authors characterize two recent isolates of Zika virus from Senegal. The authors show increased mosquito infection and potential transmission rates as well as higher virulence in a murine model of the two isolates than recent Asian/American isolates.

The results of the study are not surprising as Zika is endemic in parts of West Africa. Previously, multiple groups have performed mosquito and animal studies with an isolate of Zika DAK41525 isolated in 1984 in Senegal, and have shown higher mosquito infection rates, virulence in multiple murine models, fetal abnormalities, and sexual transmission rates in three non-human primate species (African green, rhesus, and cynomolgus). Consequently, the

these results are in agreement with the previous findings and suggest lineage specific differences.

The present study has several issues that need to be addressed with additional experiments:

1) Infectious virus need to be quantitated by plaque assay in both mosquito and murine studies. This is considerable limitation in the study.

Response: We agree with the Reviewer that infectious assays can be useful to confirm the presence of infectious virus, not just viral RNA, but this is not always necessary. Below we provide justification for when we used molecular versus infectious assays.

a) Infection rates in mosquitoes should be determined by infectious virus assays as the presence of RNA dose not correspond to productive infection.

Response: We agree with the Reviewer that viral RNA does not necessarily represent infectious virus. However, we believe that detection of viral RNA is a relevant proxy to determine mosquito infection rates, except for early time points (<7 days) when the true infection rate can be underestimated (because of low levels of viral RNA in the first few days of infection) or overestimated (due to the transient persistence of viral RNA after the blood meal even in the absence of infection). We are confident that the viral RNA detected 7 days post oral exposure and later did not simply carry over from the blood meal because we used a qualitative RT-PCR assay with a clear-cut readout that is not prone to such false positives. As shown in the picture below (provided only for the Reviewer's eyes), mosquitoes were scored based on the visually obvious presence or absence of a PCR amplicon on an electrophoresis gel.

Therefore, detection of viral RNA after day 7 post exposure does reflect a productive infection and accordingly this is a common method to determine ZIKV infection rates in *Ae. aegypti* (e.g., Diagne CT et al. BMC Infect Dis 2015; Hall-Mendelin S et al. PLoS NTDs 2016; Ciota AT et al. Emerg Infect Dis 2017; Baidaliuk A et al. J Virol 2019).

b) The presence of RNA in the saliva does not correspond to transmission of infectious virus.

Response: We apologize if it was unclear but we did detect infectious virus in mosquito saliva. The saliva samples were inoculated onto C6/36 cells to detect ZIKV by focus-forming assay (FFA). To clarify this point, we explicitly specified on line 155 and in the Fig. 2 caption that transmission efficiency was determined by FFA.

c) In both figs. 4D and 5B, infectious virus needs to be quantitated.

Response: Due to the very small size of tissue samples collected from the mouse embryos, they were only processed for the detection and quantification of viral RNA by quantitative RT-PCR. Not only does quantitative RT-qPCR require a smaller amount of input material, it is also more sensitive than infectious titration methods such as plaque assay. To complement our molecular assays, we also carried out fluorescence immunohistochemistry on various tissues of the mouse embryos, showing that staining was stronger for the Senegal_2015 strain than for the F_Polynesia_2013 strain (Fig. 5A).

To address the Reviewer's concern, we performed endpoint virus titrations for brain and testis, the two most relevant tissues for which enough biological material was still available. As shown in the figure below, viral titers were consistently and significantly higher in the brain and testis samples of AG129 mice infected with the African ZIKV strains compared to those from mice infected with the Asian ZIKV strains. These new data were added to the results section (lines 306-311) and incorporated as a new panel E in Figure 4.

2) The concentration of infectious virus concentration present immediately after oral infection to determine the ingested dose.

Response: The ingested infectious dose can be estimated as the product of the empirically determined blood meal titer and the blood meal volume. Based on a typical blood meal size of 2.5 μ l for *Ae. aegypti* (Ogunrinade A Afr J Med Med Sci 1980), the ingested infectious dose ranged from 995 to 1,577 FFU per mosquito in the high-dose experiment, and from 125 to 158 FFU per mosquito in the low-dose experiment. We provided the estimates of ingested dose on lines 162-164 and lines 198-199.

3) Earlier time points prior to 7 dpi for the mosquito studies need to be performed. In Fig. 4A, all isolates show a high rate of RNA in the mosquitoes.

Response: We did not monitor infection rate and transmission efficiency prior to day 7 because transmission rarely occurs and infection rates can be underestimated at earlier time points (Tesla B et al. PLoS NTDs 2018). We added this justification on lines 156-159.

4) The use of Asian isolates in the high and low dose mosquito study is puzzling. Five isolates are used for high dose, and the low dose utilizes only one isolate. The low dose mosquito experiment should be performed with other Asian isolates.

Response: We apologize if it was unclear but the low-dose experiment was designed to test whether the higher transmissibility of the Senegal_2015 strain in the high-dose experiment was representative of the African ZIKV lineage or specific to this strain. Therefore, the low-dose experiment included two African ZIKV strains and one Asian strain as a reference. We chose the F_Polynesia_2013 strain as the reference because it was the best-transmitted Asian strain in the high-dose experiment. We clarified this point on lines 195-196.

5) The murine studies in Fig. 4 show that Thailand and French Polynesian are the two most attenuated isolates relative to the Senegal isolates. The subsequent experiments in Fig 4C and D to determine tissue viral load and mouse embryo experiments in Fig 5 will predictably yield greater differences than the African isolates. This is a puzzling choice of Asian isolates for Figs 4 and 5. Similar is true for Fig. 6.

Response: Our overall goal was to compare the transmissibility and pathogenicity of seven low-passage ZIKV strains representing the recently circulating viral genetic diversity. Specifically, our main focus was to compare ZIKV strains from the African lineage versus the Asian lineage. We apologize if we misunderstand the Reviewer's comment, but the subset of Asian strains chosen for follow-up experiments was based on the following rationale. We chose the Thailand_2014 strain because (i) it displayed the most 'attenuated' phenotype in the survival experiment (Fig. 4B) and (ii) represented a pre-epidemic strain (i.e., not associated with a human outbreak) from the Asian lineage. We also chose the F_Polynesia_2013 strain because (i) it displayed an intermediate phenotype among other Asian strains in the survival experiment (Fig. 4B) and (ii) represented an epidemic strain from the Asian lineage. We clarified this point on lines 289-291.

Reviewer #3 (Remarks to the Author):

The manuscript reports a study on Zika virus transmission using experimental transmission trials as well as epidemiological modelling. The objectives were (i) to compare the transmissibility of Asian and African lineage, (ii) to assess the ability to disrupt embryonic development. This study covers a wide range of aspects, which allow for a better characterization of the ZIKV transmission processes at host and vector scales and pathogenicity. Finally, the authors successfully combined multiple disciplines in a comprehensive way. Although my field of expertise is related with mathematical modelling, I appreciated the whole manuscript and the global approaches the authors used. The

modelling section is also well described with the code and ad-hoc references. I only have minor comments:

- The study aims to compare African and Asian lineage in terms of transmissibility and pathogenicity. The title should reflect this comparison.

Response: Following the Reviewer's suggestion, we changed the title to "Recent African strains of Zika virus display higher transmissibility and fetal pathogenicity than Asian strains".

- Some parts of the result section would better stand (and may be somehow redundant with) Material and methods. For example, lines 157-163, or 215-231.

Response: According to the Reviewer's suggestion, we removed these parts from the results section.

- Statistical analyses (L 262-284): Data are compared for each time point. Here we have longitudinal data with non-independent samples on each individual. A global analysis on the whole trajectory using mixed effect model would allow to compare more rigorously the results. Maybe a survival analysis could have been used.

Response: We agree with the Reviewer that a mixed model accounting for the random effect of the individual mice is more appropriate to analyze repeated measures than separate analyses for each time point. Body weight and viremia levels were compared by repeated measures analysis (restricted maximum-likelihood method) using a mixed model in which ZIKV strain was nested within lineage and mouse (random effect) was nested within ZIKV strain. We revised the corresponding paragraphs in the results and methods sections on lines 258-264, lines 268-272 and lines 299-302.

- Line 296 . Why "since"? I do not see the causality here.

Response: To clarify this sentence, we changed it to "Surprisingly, using a 1,000-fold lower inoculum delayed the onset of disease by only one day for the African ZIKV strains, for which all the mice had to be euthanized on day 7 post infection.

- L484-497. The authors explain here that the absence of large outbreaks in Africa are likely due to lower susceptibility of *A. Aegypti Formosus* mostquitoes. Reference 71 (which has to be updated in the reference list) supports this fact, providing some quantitative data. Could these data be used to feed the model to assess if this assumption is sufficient to explain the absence of large outbreaks in Africa?

Response: We agree with the Reviewer that in principle, we could test our hypothesis that the lower ZIKV susceptibility of *Ae. aegypti formosus* explains the lack of large-scale epidemics in Africa by adjusting parameter estimates in the epidemiological model. Regretfully, the data available from our recent study (ref. 71 was updated to refer to the corresponding publication, which is now in press) on the differences in ZIKV susceptibility between *Ae. aegypti* subspecies do not allow a full comparison. We do have empirical mouse-to-mosquito transmission data obtained with an *Ae. aegypti formosus* population from Gabon, obtained in a side-by-side experiment with *Ae. aegypti aegypti* mosquitoes

from Guadeloupe. However, we are lacking ZIKV strain-specific transmission kinetics data to model mosquito-to-human transmission events for *Ae. aegypti formosus*.

To address this point, we used the mouse-to-mosquito transmission data obtained with the *Ae. aegypti formosus* population from Gabon (now provided in Fig. S1) to model human-to-mosquito transmission events. The simulation results (now provided in a new Fig. S2) showed a strongly reduced epidemic potential for all ZIKV strains, but it did not abolish their epidemic potential. Note that these simulations underestimate the true reduction in epidemic risk because they only account for the lower infection rate, but not for the lower transmission rate, of *Ae. aegypti formosus*.

- Line 828. The authors mentioned a non-linear regression to estimate parameters K, B and M. Some words should be given in the result section and a table for the parameters could be given.

Response: Following the Reviewer's comment, we realized that the 3-parameter regression model that we used to model the dynamics of ZIKV transmissibility by mosquitoes was not the best-fit function. We thus opted for a simpler 2-parameter log-logistic model that better captured the available data. We provided the parameter estimates of the log-logistic regression in a new Supplemental Table 2. The simulation output presented in Fig. 3 and in the results section were updated accordingly, although the conclusions remained unchanged.

Reviewer #4 (Remarks to the Author):

In this manuscript Aurby F and co-authors performed in-depth analysis of the viral fitness of Asian and African strains of ZIKV in mosquitoes and mouse model in order to understand whether the ability of the epidemic properties of the Asian strains are determined by their adaptive evolution resulting in higher virulence or better transmission. Authors found that despite much higher rate of outbreaks, Asian strains of ZIKV are less virulent and pathogenic in mice and have lower transmission capacity compared to the African strains. Authors therefore link outbreaks of Asian ZIKV to the socio-geographical factors. Even though it has been well-established that African strains have higher viral fitness than Asian strains, all previous studies used lab-adapted old African strains that were passaged multiple times in cell lines and animals, which could have increased their virulence in the laboratory models. The study of Aurby et al used recent isolates of ZIKV (Senegal 2011, 2015) that had undergone only 2 passages in C6/36 cells, resulting for the first time in thorough characterization of African ZIKV lineage which is not confounded by the lab adaptation. This brings novelty and significance to this study even though its conclusions are not different to the current knowledge.

This is a well-executed study and well-prepared manuscript with knowledgeable use of statistical and computational methods and clear data presentation, which made it a pleasure to read. However, I have a number of concerns that need to be addressed:

1. All Asian strains used in this study had higher passage number (at least 4 passages) as well as in different cells/hosts than both African strains which had only 2 passages in C6/36 cells. Therefore, the emphasis in the manuscript on low passage history for all strains needs to be toned down. The other difference between Asian and African strains is the source of the virus – all Asian strains were isolated from human serum, while both African strains were isolated from mosquitoes. This could have significant influence on viral properties. Both points need to be made clear in the text and also discussed.

Response: All Asian ZIKV strains used in this study were indeed passaged 2-3 more times in cell culture than the two African strains. Although a clear definition of ‘low-passage’ virus does not exist, we consider that using this term for all our ZIKV strains is consistent with the current literature. The term low-passage has been used in previous publications to qualify several of our Asian ZIKV strains: Cambodia_2010 (Rossi SL et al. Am J Trop Med Hyg 2016), F_Polynesia_2013 (Atieh T et al. Sci Rep 2016), Philippines_2012 and Thailand_2014 (Smith DR et al. Am J Trop Med Hyg 2018). Likewise, the term low-passage has been used to qualify African ZIKV strains that were passaged 5 (41662-DAK; Atieh T et al. Sci Rep 2016) and 7 times (DakAr41524; Duggal NK et al. Am J Trop Med Hyg 2017) in cell culture.

We cannot conclusively rule out that the small difference in passage number between the African and the Asian ZIKV strains of our panel could have influenced the results of our study. Nevertheless, there is accumulating evidence that African ZIKV strains are more pathogenic and more transmissible than Asian strains, regardless of their passage history. In fact, our study demonstrates that this is the case even when African strains have minimal passage history, unlike numerous earlier studies that used heavily passaged strains from the African lineage. It is also unlikely that the source of the virus (mosquito pool or human serum) alone, could explain the observed differences in transmissibility and pathogenicity. Indeed, we performed a pilot experiment (not included in the manuscript) with an Asian ZIKV strain (ZIKV MEX 2-81; GenBank accession number KX446950) that was originally isolated from mosquitoes. Similar to the ZIKV Thailand_2014 strain, the ZIKV MEX 2-81 strain showed an attenuated phenotype in the AG129 mouse model (using an 1000-PFU inoculum), as shown in the figure below, provided only for the Reviewer’s eyes.

Thus, the original source of the virus does not seem to be a primary determinant of the viral properties. Of note, all of our ZIKV stocks were produced in C6/36 mosquito cells prior to

performing the experiments, therefore the final 'host' was effectively the same. We added a sentence to address this point in the discussion section on lines 472-475.

2. As a model of viral pathogenesis in mammalian host, authors used AG129 mice that are highly immunocompromised and embryos of WT mice, which have not yet fully developed the immune system. Therefore, if epidemic properties of the Asian viral strains are determined by better immune suppression or evasion, this cannot be elucidated in the utilised models. I appreciate that fully immunocompetent animal models of ZIKV infection are not available, but believe that authors should discuss this limitation in the manuscript.

Response: We primarily focused on the intrinsic potential of ZIKV strains to replicate to higher levels and/or cause more pathogenesis. Although immunocompromised mouse models have their shortcomings, they have also proved their worth. Models using immunocompromised mice better recapitulate the neurotropic nature of ZIKV than immunocompetent mice, which are generally resistant to ZIKV infection and experience only transient viremia. We pointed this out in the discussion section on lines 432-436. In addition, the pivotal role of interferon in restricting ZIKV infections was identified by using knockout mice genetically modified for different components of the antiviral signaling pathway, including A129, AG129, *Ifnar1*^{-/-}, and *Stat2*^{-/-} mice (Rossi SL et al. Cell Stem Cell 2016; Tripathi S et al. PLoS Pathog 2017). Indeed, ZIKV has the potential to antagonize innate immune responses of the host, which may involve various ZIKV proteins (Bowen JR, et al. PLoS Pathog 2017; Grant A et al. Cell Host Microbe 2016; Kumar A et al. EMBO Rep 2016; Wu Y et al. Cell Discov 2017; Xia H et al. Nat Commun 2018). The mechanism by which the antagonistic effect is brought about may be shared by ZIKV strains from both lineages (Seong RK et al. Pathogens 2020; Xia H et al. Nat Commun 2018), may be strain- or lineage-dependent (Österlund et al. Sci Rep 2019; Tripathi S et al. PLoS Pathog 2017), or has only been described for specific ZIKV strains (Wu Y et al. Cell Discov 2017). Models employing immunocompromised mice are less/not suitable for comparing the ability of ZIKV strains to suppress or evade the host's immune system, which may additionally contribute to their epidemic potential.

We added a paragraph to the discussion section (lines 444-451) to acknowledge the limitations of immunocompromised mouse models for elucidating immunological mechanisms possibly associated with epidemic potential.

3. Given that all strains were fully sequenced in this study I was surprised not to see any mentioning of which changes may be related to the differences in viral properties. Surely this would be of significant interest to the authors themselves and also to the research community. At least a Table presenting key changes between the strains and some discussion on potential role of them needs to be included. I would also expect to see some data on how at least the key differences in the sequence influence viral properties. Infectious clones are now available for various ZIKV strains and could be used to answer these questions.

Response: We agree with the Reviewer that identifying nucleotide variants in the viral genome that are associated with phenotypic differences is highly desirable. However, this is

hindered by at least two major difficulties: (1) The African and Asian ZIKV lineages are highly divergent. Their nucleotide divergence is ~12%, which translates into more than 100 different amino-acid residues across the open reading frame (Smith DR et al. AJTMH 2018). In contrast, for instance, the American ZIKV strains only differ from the rest of the Asian strains at less than ~30 amino-acid residues. (2) The phenotypic differences likely result from complex combinations of genetic variants. Fitness differences between viral lineages often result from epistatic relationships, as was observed for dengue (Syenina A et al. PNAS 2020) and chikungunya (Tsetsarkin KA et al. PNAS 2011) viruses. The large number of possible genetic combinations between African and Asian ZIKV lineages makes them less tractable by conventional reverse genetics using infectious clones.

To address this point, we added a paragraph to the discussion section (lines 553-561) and referred to an earlier study that inventoried the substitutions that occurred during ZIKV emergence in the Pacific and Latin America using the African prototypic ZIKV strain MR766 as a reference (Pettersson JHO et al. mBio 2016).

Reviewers' Comments:

Reviewer #2:

Remarks to the Author:

Majority of the previous comments have not been addressed experimentally.

1) Previous research by multiple groups have shown that West African isolates displayed higher virulence both in vitro and in vivo. Consequently, recent isolates from the same region presented in this manuscript confirm previous findings. Nothing in the research is novel or surprising.

2) Characterization of virus isolates in mosquito and murine models belongs in more technical journals. This manuscript is very similar to other similar Zika studies in lower impact technical journals.

3) No molecular clones are utilized.

4) Experimental design has flaws.

a) Infectious virus should be measured along with RNA. RNA does not necessarily translate to infectious virus.

b) Lack of uniformity in mosquito experiments i.e. Five isolates are used for high dose, and the low dose utilizes only one isolate.

c) The murine studies in Fig. 4 use two most attenuated isolates (Thailand and French Polynesian) and will yield greater differences than the African isolates.

Reviewer #3:

Remarks to the Author:

The authors addressed the different concerns on mathematical and statistical modelling. In view of their clarifications and modifications, I do not have any further comments.

Reviewer #4:

Remarks to the Author:

All my comments have been adequately addressed